# Metric Flow Matching for Smooth Interpolations on the Data Manifold

**Kacper Kapuśniak**[1][*], **Peter Potaptchik**[1], **Teodora Reu**[1], **Leo Zhang**[1],
**Alexander Tong**[2,3], **Michael Bronstein**[1,4], **Avishek Joey Bose**[1,2], **Francesco Di Giovanni**[1]

[1]University of Oxford, [2]Mila, [3]Université de Montréal, [4]AITHYRA

## Abstract

Matching objectives underpin the success of modern generative models and rely on constructing conditional paths that transform a source distribution into a target distribution. Despite being a fundamental building block, conditional paths have been designed principally under the assumption of *Euclidean geometry*, resulting in straight interpolations. However, this can be particularly restrictive for tasks such as trajectory inference, where straight paths might lie outside the data manifold, thus failing to capture the underlying dynamics giving rise to the observed marginals. In this paper, we propose METRIC FLOW MATCHING (MFM), a novel simulation-free framework for conditional flow matching where interpolants are approximate geodesics learned by minimizing the kinetic energy of a data-induced Riemannian metric. This way, the generative model matches vector fields on the data manifold, which corresponds to lower uncertainty and more meaningful interpolations. We prescribe general metrics to instantiate MFM, independent of the task, and test it on a suite of challenging problems including LiDAR navigation, unpaired image translation, and modeling cellular dynamics. We observe that MFM outperforms the Euclidean baselines, particularly achieving SOTA on single-cell trajectory prediction.

## 1 Introduction

A central task in many natural and scientific domains entails the inference of system dynamics of an underlying (physical) process from noisy measurements. A core challenge, in these application domains such as biomedical ones—e.g. tracking health metrics [Oeppen and Vaupel, 2002] or diseases [Hay et al., 2021]—is that one typically lacks access to entire time-trajectories and can only leverage cross-sectional samples. An even more poignant example is the case of single-cell RNA sequencing [Macosko et al., 2015, Klein et al., 2015], where measurements are *sparse* and *static*, due to the procedure being expensive and destructive. Consequently, the nature of these tasks demands the design of frameworks capable of reconstructing the temporal dynamics of a system (e.g. cells) from observed time marginals that contain finite samples. This overarching problem specification is referred to as **trajectory inference** [Hashimoto et al., 2016, Lavenant et al., 2021].

To address this challenge, we rely on matching objectives, a powerful generative modeling paradigm encompassing successful approaches including diffusion models [Sohl-Dickstein et al., 2015, Song et al., 2021], flow matching [Lipman et al., 2023, Liu et al., 2022, Albergo and Vanden-Eijnden, 2022], and finding a Schrödinger Bridge [Schrödinger, 1932, Léonard, 2013]. Specifically, to reconstruct the unknown dynamics $t \mapsto p_t^*$ between observed time marginals $p_0$ and $p_1$, we leverage Conditional Flow Matching (CFM) [Tong et al., 2023b], a simulation-free framework which constructs a probability path $p_t$ through interpolants $x_t$ connecting samples of $p_0$ to samples of $p_1$. In general, $x_t$ is designed under the assumption of *Euclidean geometry*, resulting in *straight*

---

[*]Corresponding author: `kacper.kapusniak@cs.ox.ac.uk`
Code is available at https://github.com/kksniak/metric-flow-matching

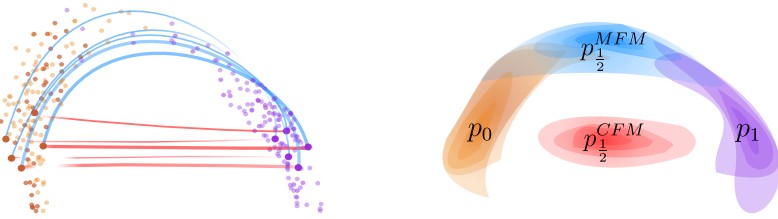

Figure 1: In orange and violet the source and target distributions. On the left, straight interpolations vs interpolations following a data-dependent Riemannian metric. On the right, densities of reconstructed marginals at time $t = \frac{1}{2}$, using Conditional Flow Matching and METRIC FLOW MATCHING (MFM), respectively. MFM provides a more meaningful reconstruction supported on the data manifold.

trajectories. However, in light of the widely accepted "manifold hypothesis" [Tenenbaum et al., 2000, Belkin and Niyogi, 2003], the target time-evolving density $p_t^*$ is supported on a *curved* low-dimensional manifold $\mathcal{M} \subset \mathbb{R}^d$—a condition satisfied by cells in the space of gene expressions [Moon et al., 2018]. As such, straight interpolants stray away from the data manifold $\mathcal{M}$, leading to reconstructions $p_t$ that fail to model the nonlinear dynamics generated by the underlying process.

**Present work**. We aim to design interpolants $x_t$ that stay on the data manifold $\mathcal{M}$ associated with the underlying dynamics. Nonetheless, parameterizing the lower-dimensional manifold $\mathcal{M}$ is prone to instabilities [Loaiza-Ganem et al., 2022] and may require multiple coordinate systems [Salmona et al., 2022]. Accordingly, we adopt the "metric learning" approach [Xing et al., 2002, Hauberg et al., 2012], where we still work in the ambient space $\mathbb{R}^d$, but equip it with a *data-dependent Riemannian metric* $g$ whose shortest-paths (geodesics) stay close to the data points, and hence to $\mathcal{M}$ [Arvanitidis et al., 2021]. We introduce METRIC FLOW MATCHING (MFM), a simulation-free generalization of CFM where interpolants $x_t$ are *learned* by minimizing a geodesic loss that penalizes the velocity measured by the metric $g$. As a result, $x_t$ approximates the geodesics of $g$ and hence tends towards the data, leading to the evaluation of the matching objective in regions of lower uncertainty. Therefore, the resulting probability path $p_t$ is supported on the data manifold for all $t \in [0, 1]$, giving rise to a more natural reconstruction of the underlying dynamics in the trajectory inference task, as depicted in Figure 1.

Our **main contributions** are:

(1) We prove that given a dataset $\mathcal{D} \subset \mathbb{R}^d$, one can always construct a metric $g$ on $\mathbb{R}^d$ such that the geodesics connecting $x_0$ sampled from $p_0$ to $x_1$ sampled from $p_1$, always lie close to $\mathcal{D}$ (§3.1).

(2) We propose METRIC FLOW MATCHING, a novel framework generalizing CFM to the Riemannian manifold associated with any data-dependent metric $g$. In MFM, before training the matching objective, we learn interpolants that stay close to the data by minimizing a cost induced by $g$ (§3). MFM is simulation-free and stays relevant when geodesics lack closed form and hence Riemannian Flow Matching [Chen and Lipman, 2023] is not easily applicable (§3.2).

(3) We prescribe a universal way of designing a data-dependent metric, independent of the specifics of the task, which enforces interpolants $x_t$ to stay supported on the data manifold (§4.1). Through the proposed metric, $x_t$ depends on the entire data manifold and not just the endpoints $x_0$ and $x_1$ sampled from the marginals. By accounting for the Riemannian geometry induced by the data, MFM generalizes existing approaches that construct $x_t$ by minimizing energies (§4.2).

(4) We propose OT-MFM, an instance of MFM that relies on Optimal Transport to draw samples from the marginals (§4). Empirically, we show that OT-MFM attains SOTA results for reconstructing single-cell dynamics (§5). Additionally, we validate the versatility of the framework through tasks such as 3D navigation with LiDAR point clouds and unpaired translation of images.

## 2 Preliminaries and Setting

We review Conditional Flow Matching, which forms the basis of METRIC FLOW MATCHING §3. Next, we recall basic notions of Riemannian geometry, with emphasis on constructing geodesics.

**Notation and convention**. We let $\mathbb{R}^d$ be the ambient space where data points are embedded. A random variable $x$ with a distribution $p$ is denoted as $x \sim p(x)$. A function $\varphi$ depending on time $t$, space $x$ and learnable parameters $\theta$, is denoted by $\varphi_{t,\theta}(x)$, and its time derivative by $\dot{\varphi}_{t,\theta}$. We also let $\delta_{x_0}(x)$ be the Dirac function centered at $x_0$ and assume that all distributions are absolutely continuous, which allows

us to use densities. We denote the space of symmetric, positive definite $d \times d$ matrices as $\mathrm{SPD}(d)$, and let $\mathbf{G}(x) \in \mathrm{SPD}(d)$ be the coordinate representation of a Riemannian metric at some point $x$.

**Conditional Flow Matching**. We consider a source distribution $p_0$ and a target distribution $p_1$ defined on $\mathbb{R}^d$. We are interested in finding a map $f$ that pushes forward $p_0$ to $p_1$, i.e. $f_\# p_0 = p_1$. In line with Continuous Normalizing Flows [Chen et al., 2018], we look for a map of the form $f = \psi_1$, where the time-dependent diffeomorphism $\psi_t : \mathbb{R}^d \to \mathbb{R}^d$ is the flow generated by a vector field $u_t$, i.e. $d\psi_t(x)/dt = u_t(\psi_t(x))$, with $\psi_0(x) = x$, for all $x \in \mathbb{R}^d$. If we define the density path $p_t = [\psi_t]_\# p_0$, then $p_t$ and $u_t$ satisfy the continuity equation and we say that $p_t$ is *generated* by $u_t$.

If density path $p_t$ and vector field $u_t$ are known, we could regress a vector field $v_{t,\theta}$, modeled as a neural network, to $u_t$ by minimizing $\mathcal{L}_{\mathrm{FM}}(\theta) = \mathbb{E}_{t,p_t(x)} \|v_{t,\theta}(x) - u_t(x)\|^2$. Since $p_t$ and $u_t$ are typically intractable though, Conditional Flow Matching (CFM) [Lipman et al., 2023, Albergo and Vanden-Eijnden, 2022, Liu et al., 2022] simplifies the problem by assuming that $p_t$ is a mixture of conditional paths: $p_t(x) = \int p_t(x|z)q(z)dz$, where $z = (x_0, x_1)$ is sampled from a joint distribution $q$ with marginals $p_0$ and $p_1$, and $p_t(x|z)$ satisfy $p_0(x|z) \approx \delta_{x_0}(x)$ and $p_1(x|z) \approx \delta_{x_1}(x)$. If $\psi_t(x|x_0, x_1)$ denotes the flow generating $p_t(x|z)$, then the CFM objective is

$$\mathcal{L}_{\mathrm{CFM}}(\theta) = \mathbb{E}_{t,(x_0,x_1)\sim q} \|v_{t,\theta}(x_t) - \dot{x}_t\|^2, \tag{1}$$

where $x_t := \psi_t(x_0|x_0, x_1)$ are the *interpolants* from $x_0$ to $x_1$. Since $\mathcal{L}_{\mathrm{FM}}$ and $\mathcal{L}_{\mathrm{CFM}}$ have same gradients [Lipman et al., 2023, Tong et al., 2023b], we can use the tractable conditional paths to learn $v_\theta$. As in §3 we design $x_t$ to approximate geodesics, we review key notions from Riemannian geometry.

**Riemannian manifolds**. A Riemannian manifold $(\mathcal{M}, g)$ is a smooth orientable manifold $\mathcal{M}$ equipped with a smooth map $g$ assigning to each point $x$ an inner product $\langle \cdot, \cdot \rangle_g$ defined on the tangent space of $\mathcal{M}$ at $x$. We let $\mathbf{G}(x) \in \mathrm{SPD}(d)$ be the matrix representing $g$ in coordinates, at any point $x$, with $d$ the dimension of $\mathcal{M}$. Integration is taken with respect to the volume form $d\mathrm{vol}$ (see Appendix §A for details). A continuous, positive function $p$ is then a probability density on $(\mathcal{M}, g)$, i.e. $p \in \mathbb{P}(\mathcal{M})$, if $\int p(x)d\mathrm{vol}(x) = 1$. Naturally, it is possible to define curves $\gamma_t$, indexed by time $t$. A *geodesic* is then the curve $\gamma_t^*$ that minimizes the distance with respect to $g$. Specifically, the geodesic connecting $x_0$ to $x_1$ in $\mathcal{M}$, can be found by minimizing the length functional:

$$\gamma_t^* = \underset{\gamma_t : \gamma_0 = x_0, \gamma_1 = x_1}{\arg\min} \int_0^1 \|\dot{\gamma}_t\|_{g(\gamma_t)} \, dt, \quad \|\dot{\gamma}_t\|_{g(\gamma_t)} := \sqrt{\langle \dot{\gamma}_t, \mathbf{G}(\gamma_t)\dot{\gamma}_t \rangle}, \tag{2}$$

where $\dot{\gamma}_t$ is velocity. From eq. (2), we see that geodesics tend towards regions where $\|\mathbf{G}(x)\|$ is small. In Euclidean geometry (i.e., $\mathcal{M} = \mathbb{R}^d$ and $\mathbf{G}(x) = \mathbf{I}$), $\gamma_t^*$ is a straight line with constant speed.

## 3 Metric Flow Matching

We introduce METRIC FLOW MATCHING (MFM), a new simulation-free framework that generalizes CFM by constructing probability paths supported on the data manifold. MFM learns interpolants $x_t$ in eq. (1) whose velocity minimizes a data-dependent Riemannian metric assigning a lower cost to regions with high data concentration. Consequently, the CFM objective is evaluated in areas of low data uncertainty, and the corresponding vector field, $v_\theta$ in eq. (1), learns to pass through these regions.

We structure this section as follows. In §3.1 we discuss the trajectory inference problem and how straight interpolants $x_t$ in CFM result in undesirable probability paths whose support is not defined on the data manifold. We remedy this problem by choosing to represent the data manifold via a Riemannian metric in $\mathbb{R}^d$, such that geodesics avoid straying away from the samples in the training set. In §3.2 we introduce MFM and compare it with Riemannian Flow Matching [Chen and Lipman, 2023].

### 3.1 Metric learning

Assume that $p_0$ and $p_1$ are empirical distributions and that we have access to a dataset of samples $\mathcal{D} = \{x_i\}_{i=1}^N$—in practice $\mathcal{D}$ is constructed concatenating samples from both the source and target distributions. We are interested in the problem of **trajectory inference** [Lavenant et al., 2021], where we need to reconstruct an unknown dynamics $t \mapsto p_t^*$, with observed time marginals $p_0^* = p_0$ and $p_1^* = p_1$—the extension to multiple timepoints is easy. In many realistic settings, including single-cell RNA sequencing [Macosko et al., 2015], time measurements are sparse, and leveraging biases from

the data is hence key to achieving a faithful reconstruction. For this reason, we invoke the "manifold hypothesis" [Bengio et al., 2013], where the data arises from a low-dimensional manifold $\mathcal{M} \subset \mathbb{R}^d$—a property satisfied by cells embedded in the space of gene expressions [Moon et al., 2018]:

$$\operatorname{supp}(p_t^*) := \{x \in \mathbb{R}^d : p_t^*(x) > 0\} \subset \mathcal{M}, \quad t \geq 0. \tag{3}$$

Since any regular time dynamics can be described using the continuity equation generated by some vector field $v_t^*$ [Ambrosio et al., 2005, Theorem 8.3.1], [Neklyudov et al., 2023a], we rely on CFM, and approximate $p_t^*$ via the probability path $p_t$ associated with $v_{t,\theta}$ in eq. (1). From eq. (3), it follows that a valid reconstruction entails $\operatorname{supp}(p_t) \subset \mathcal{M}$. As the support of $p_t$ is determined by the interpolants $x_t$ in eq. (1), we need $x_t$ to be constrained to stay on $\mathcal{M}$. However, in the classical CFM setup, this condition is violated since interpolants are often *straight* lines, *agnostic* of the data's support: $x_t = tx_1 + (1-t)x_0$ [Tong et al., 2023b, Shaul et al., 2023], with $x_0, x_1$ sampled from the marginals. In this case, if $p_t(x|x_0, x_1) \approx \delta_{x_t}(x)$, then the support of $p_t$ satisfies

$$\operatorname{supp}(p_t) \subset \{y \in \mathbb{R}^d : \exists\ (x_0, x_1) \sim q :\ y = tx_1 + (1-t)x_0\}.$$

However, the dynamics $t \mapsto p_t^*$ is often *nonlinear*, as for single-cells [Moon et al., 2018], meaning that $\operatorname{supp}(p_t) \not\subseteq \mathcal{M}$. Straight interpolants are hence too restrictive and should instead be supported on $\mathcal{M}$ so to replicate *actual* trajectories from $x_0$ to $x_1$, which are generated by the underlying process.

Operating on a lower-dimensional $\mathcal{M}$ is challenging though, since it requires different coordinate systems [Schonsheck et al., 2019, Salmona et al., 2022] and may incur overfitting [Loaiza-Ganem et al., 2022, 2024]. Nonetheless, a key property posited by the "manifold hypothesis" is that $\mathcal{M}$ concentrates around the data points $\mathcal{D}$ [Arvanitidis et al., 2022, Chadebec and Allassonnière, 2022]. As such, interpolants $x_t$ should remain close to $\mathcal{D}$. Therefore, instead of changing the dimension, we design $x_t$ to minimize a cost in $\mathbb{R}^d$ that is lower on regions close to $\mathcal{D}$. We achieve this following the "metric learning" approach, [Hauberg et al., 2012] where we equip $\mathbb{R}^d$ with a suitable Riemannian metric $g$.

**Definition 1.** *A data-dependent metric $g$ on $\mathbb{R}^d$ is a smooth map $g : \mathbb{R}^d \to \operatorname{SPD}(d)$ parameterized by the dataset $\mathcal{D} = \{x_i\}_{i=1}^N \subset \mathbb{R}^d$, i.e. $g(x) = \mathbf{G}(x; \mathcal{D}) \in \operatorname{SPD}(d), \forall x \in \mathbb{R}^d$.*

We describe a specific metric $g$ in §4, and note that $g$ can also enforce constraints from the task (§7). Naturally, if $\mathbf{G}(x; \mathcal{D}) = \mathbf{I}$, we recover the Euclidean metric. We show that we can always construct $g$ so that the geodesics $\gamma_t^*$ stay close to the data $\mathcal{D}$. For details, we refer to Theorem B.1 in Appendix §B.

**Proposition 1** (Informal). *Given a dataset $\mathcal{D} \subset \mathbb{R}^d$, let $g$ be any metric such that: (i) The eigenvalues of $\mathbf{G}(x; \mathcal{D})$ do not approach zero when $x$ is distant from $\mathcal{D}$; (ii) $\|\mathbf{G}(x; \mathcal{D})\|$ is sufficiently small if $x$ is close to $\mathcal{D}$. Then for each $(x_0, x_1) \sim q$, the geodesic $\gamma_t^*$ connecting $x_0$ to $x_1$ stays close to $\mathcal{D}$.*

Given $g$ as in the statement, if $x_t = \gamma_t^*$ and $p_t(x|x_0, x_1) \approx \delta_{x_t}(x)$, then the probability path $p_t$ generated by $v_\theta$ in eq. (1) has support near $\mathcal{D}$, i.e. $\operatorname{supp}(p_t)$ lies close to the data manifold $\mathcal{M}$, which is our goal. Unfortunately, for all but the most trivial metrics $g$ on $\mathbb{R}^d$, it is not possible to obtain closed-form expressions for the geodesics $\gamma_t^*$. As such, finding the geodesic $\gamma_t^*$ necessitates the expensive *simulation* of second-order nonlinear Euler-Lagrange equations [Hennig and Hauberg, 2014].

### 3.2 Parameterization and optimization of interpolants

Consider a metric $g$ on $\mathbb{R}^d$ as in Definition 1 whose geodesics $\gamma_t^*$ lie close to $\mathcal{D}$ as per Proposition 1. We propose a *simulation-free* approximation to paths $\gamma_t^*$ by introducing interpolants of the form

$$x_{t,\eta} = (1-t)x_0 + tx_1 + t(1-t)\varphi_{t,\eta}(x_0, x_1), \tag{4}$$

where $\eta$ are the parameters of a neural network $\varphi_{t,\eta}$ acting as a nonlinear "correction" for straight interpolants. Note that the boundary conditions are met, i.e. that the path $x_{t,\eta}$ recovers both $x_0$ and $x_1$ at times $t=0$ and $t=1$, respectively. In fact, $x_{t,\eta}$ reduces to the convex combination between $x_0$ and $x_1$ if $\varphi_{t,\eta} = 0$, meaning that $x_{t,\eta}$ strictly generalize the straight paths used in [Lipman et al., 2023, Liu et al., 2022]. Towards the goal of learning $\eta$ so that $x_{t,\eta}$ approximates the geodesic $\gamma_t^*$, we note that $\gamma_t^*$ can be characterized as the path minimizing the convex functional $\mathcal{E}_g$ below:

$$\gamma_t^* = \underset{\gamma_t:\, \gamma_0 = x_0,\, \gamma_1 = x_1}{\arg\min} \mathcal{E}_g(\gamma_t), \quad \mathcal{E}_g(\gamma_t) := \mathbb{E}_t\left[\dot{\gamma}_t^\top \mathbf{G}(\gamma_t; \mathcal{D})\dot{\gamma}_t\right]. \tag{5}$$

Since $\gamma_t^*$ minimizes $\mathcal{E}_g$ over all paths connecting $x_0$ to $x_1$, and $x_{t,\eta}$ in eq. (4) satisfies these boundary conditions, we can estimate $\eta$ by simply minimizing the convex functional $\mathcal{E}_g$ over $x_{t,\eta}$, which leads to the following geodesic objective (the training procedure is reported in Algorithm 1):

---

**Algorithm 1** Pseudocode for training of geodesic interpolants

---

**Require:** coupling $q$, initialized network $\varphi_{t,\eta}$, data-dependent metric $\mathbf{G}(\cdot; \mathcal{D})$
1: **while** Training **do**
2:     Sample $(x_0, x_1) \sim q$ and $t \sim \mathcal{U}(0, 1)$
3:     $x_{t,\eta} \leftarrow (1-t)x_0 + tx_1 + t(1-t)\varphi_{t,\eta}(x_0, x_1)$                 $\triangleright$ *eq.* (4)
4:     $\dot{x}_{t,\eta} \leftarrow x_1 - x_0 + t(1-t)\dot{\varphi}_{t,\eta}(x_0, x_1) + (1-2t)\varphi_{t,\eta}(x_0, x_1)$
5:     $\ell(\eta) \leftarrow (\dot{x}_{t,\eta})^\top \mathbf{G}(x_{t,\eta}; \mathcal{D})\dot{x}_{t,\eta}$      $\triangleright$ *Estimate of objective $\mathcal{L}_g(\eta)$ from eq.* (6)
6:     Update $\eta$ using gradient $\nabla_\eta \ell(\eta)$
    **return** (approximate) geodesic interpolants parametrized by $\varphi_{t,\eta}$

---

$$\mathcal{L}_g(\eta) := \mathbb{E}_{(x_0,x_1)\sim q}\left[\mathcal{E}_g(x_{t,\eta})\right] = \mathbb{E}_{(x_0,x_1)\sim q,t}\left[(\dot{x}_{t,\eta})^\top \mathbf{G}(x_{t,\eta}; \mathcal{D})\dot{x}_{t,\eta}\right]. \tag{6}$$

Given interpolants that approximate $\gamma_t^*$ and hence stay close to the data manifold, we can then rely on the CFM objective in eq. (1) to regress the vector field $v_\theta$. Since $g$ makes the ambient space $\mathbb{R}^d$ into a Riemannian manifold $(\mathbb{R}^d, g)$, we need to replace the norm $\|\cdot\|$ in eq. (1) with the one $\|\cdot\|_g$ induced by the metric, and rescale the marginals $p_0, p_1$ using the volume form induced by $g$, so to extend $p_0, p_1$ to densities in $\mathbb{P}(\mathbb{R}^d, g)$ (see Appendix §A). Similar arguments work for the joint distribution $q$. We can finally introduce our framework METRIC FLOW MATCHING that generalizes Conditional Flow Matching (1) to leverage *any* data-dependent metric $g$, by using interpolants $x_{t,\eta}$ (4), whose parameters $\eta$ are obtained from minimizing the geodesic cost $\mathcal{E}_g$. The MFM objective can be stated as:

$$\mathcal{L}_{\text{MFM}}(\theta) = \mathbb{E}_{t,(x_0,x_1)\sim q}\left[\|v_{t,\theta}(x_{t,\eta^*}) - \dot{x}_{t,\eta^*}\|^2_{g(x_{t,\eta^*})}\right], \quad \eta^* = \arg\min_\eta \mathcal{L}_g(\eta). \tag{7}$$

A description of METRIC FLOW MATCHING is given in Algorithm 2. As the interpolants $x_{t,\eta}$ approximate geodesics of $g$, in MFM the vector field $v_{t,\theta}$ is regressed on the data manifold $\mathcal{M}$, where the underlying dynamics $p_t^*$ is supported, resulting in better reconstructions. Crucially, eq. (6) only depends on time derivatives of $x_{t,\eta}$. Therefore, MFM avoids simulations and simply requires training an additional (smaller) network $\varphi_{t,\eta}$ in eq. (4), which can be done *prior* to training $v_{t,\theta}$.

**MFM versus Riemannian Flow Matching**. While CFM has already been extended to Riemannian manifolds in the Riemannian Flow Matching (RFM) framework of Chen and Lipman [2023], MFM crucially differs from RFM in two ways. To begin with, MFM relies on the data or task inducing a Riemannian metric on the ambient space which is then accounted for in the matching objective. This is in sharp contrast to RFM, which instead assumes that the metric of the ambient space is *given* and is *independent* of the data points. Secondly, RFM does not incorporate conditional paths that are learned. In fact, in the scenario above where $g$ is a metric whose geodesics $\gamma_t^*$ stay close to the data support, adopting RFM would entail replacing the MFM objective $\mathcal{L}_{\text{MFM}}$ in (7) with

$$\mathcal{L}_{\text{RFM}}(\theta) = \mathbb{E}_{t,(x_0,x_1)\sim q}\|v_{t,\theta}(\gamma_t^*) - \dot{\gamma}_t^*\|^2_{g(\gamma_t^*)}. \tag{8}$$

However, as argued above, for almost any metric $g$ on $\mathbb{R}^d$, geodesics $\gamma_t^*$ can only be found via *simulations*, which in high dimensions inhibits the easy application of RFM. Conversely, MFM designs interpolants that minimize a geodesic cost (6) and hence approximate $\gamma_t^*$ without incurring simulations.

## 4 Learning Riemannian Metrics in Ambient Space

In this section, we focus on a concrete choice of $g$, which can easily be used within MFM (§4.1). We also introduce OT-MFM, a variant of MFM that leverages Optimal Transport to find a coupling $q$ between $p_0$ and $p_1$. Next, in §4.2 we discuss how MFM generalizes recent works that find interpolants that minimize energies by accounting for the Riemannian geometry induced by the data.

### 4.1 A family of diagonal metrics: LAND and RBF

We consider a family of metrics $g_{\text{LAND}}$ as in Definition 1, independent of specifics of the data type or task. For the ease of exposition, we omit to write the explicit dependence on the dataset $\mathcal{D} = \{x_i\}_{i=1}^N$.

Given $\varepsilon > 0$, we let $x \mapsto g_{\text{LAND}}(x) \equiv \mathbf{G}_\varepsilon(x) = (\text{diag}(\mathbf{h}(x)) + \varepsilon \mathbf{I})^{-1}$ be the "LAND" metric, where

$$h_\alpha(x) = \sum_{i=1}^{N} (x_i^\alpha - x^\alpha)^2 \exp\Big( - \frac{\|x - x_i\|^2}{2\sigma^2} \Big), \quad 1 \le \alpha \le d, \tag{9}$$

with $\sigma$ the kernel size. We emphasize that $\mathbf{G}_\varepsilon(x)$ was introduced by Arvanitidis et al. [2016]—from which we borrow the name—but its algorithmic use in MFM is fundamentally different. In line with Proposition 1, we see that $\|\mathbf{G}_\varepsilon(x)\|$ is larger away from $\mathcal{D}$, thus pushing geodesics (2) to stay close to the data support, as desired. While $g_{\text{LAND}}$ is flexible and directly accounts for all the samples in $\mathcal{D}$, in high-dimension selecting $\sigma$ in eq. (9) can be challenging. For these reasons, we follow Arvanitidis et al. [2021] and introduce a variation of $g_{\text{LAND}}$ of the form $\mathbf{G}_{\text{RBF}}(x) = (\text{diag}(\tilde{\mathbf{h}}(x)) + \varepsilon \mathbf{I})^{-1}$, where

$$\tilde{h}_\alpha(x) = \sum_{k=1}^{K} \omega_{\alpha,k}(x) \exp\Big( - \frac{\lambda_{\alpha,k}}{2} \|x - \hat{x}_k\|^2 \Big), \quad 1 \le \alpha \le d, \tag{10}$$

with $K$ the number of clusters with centers $\hat{x}_k$ and $\lambda_{\alpha,k}$ the bandwidth of cluster $k$ for channel $\alpha$ (see Appendix §C for details). In particular, $h_\alpha$ is realized via a Radial Basis Function (RBF) network [Que and Belkin, 2016], where $\omega_{\alpha,k} > 0$ are *learned* to enforce the behavior $h_\alpha(x_i) \approx 1$ for each data point $x_i$ so that the resulting metric $g_{\text{RBF}}$ assigns lower cost to regions close to the centers $\hat{x}_k$. In our experiments, we then rely on $g_{\text{LAND}}$ in low dimensions, and instead use $g_{\text{RBF}}$ in high dimensions. We also note that all metrics considered are diagonal, which makes MFM more efficient. Explicitly, given the interpolants in eq. (4), the geometric loss in eq. (6) with respect to $g_{\text{RBF}}$ can be written as:

$$\mathcal{L}_{g_{\text{RBF}}}(\eta) = \mathbb{E}_{(x_0,x_1)\sim q}[\mathcal{E}_{g_{\text{RBF}}}(x_{t,\eta})] = \mathbb{E}_{t,(x_0,x_1)\sim q}\left[ \sum_{\alpha=1}^{d} \frac{(\dot{x}_{t,\eta})_\alpha^2}{\tilde{h}_\alpha(x_{t,\eta}) + \varepsilon} \right]. \tag{11}$$

We see that the loss acts as a geometric regularization, with the velocity $\dot{x}_{t,\eta}$ penalized more in regions away from the support of the dataset $\mathcal{D}$, i.e. when $\lambda_{\alpha,k}\|x_{t,\eta} - \hat{x}_k\|$ is large for all centers $\hat{x}_k$ in eq. (10).

**OT-MFM**. In eq. (7), samples $x_0, x_1$ follow a joint distribution $q$, with marginals $p_0$ and $p_1$. Since we are interested in the problem of trajectory inference, with emphasis on single-cell applications where the principle of least action holds [Schiebinger, 2021], we focus on a coupling $q$ that minimizes the distance in probability space between the source and target distributions. Namely, we consider the case where $q$ is the 2-Wasserstein optimal transport plan $\pi^*$ from $p_0$ to $p_1$ [Villani et al., 2009]:

$$\pi^* = \arg\min_{\pi \in \Pi} \int_{\mathbb{R}^d \times \mathbb{R}^d} c^2(x,y) d\pi(x,y), \tag{12}$$

where $\Pi$ are the probability measures on $\mathbb{R}^d \times \mathbb{R}^d$ with marginals $p_0$ and $p_1$ and $c$ is any cost. While we could choose $c$ based on $g_{\text{RBF}}$, we instead select $c$ to be the $L_2$ distance in $\mathbb{R}^d$ to avoid additional computations and so we can study the role played by $x_{t,\eta}$ even when $q = \pi^*$ is agnostic of the data manifold. We then propose the OT-MFM objective, where $\eta^*$ minimizes the geodesic loss in eq. (11):

$$\mathcal{L}_{\text{OT-MFM}_{\text{RBF}}}(\theta) = \mathbb{E}_{t,(x_0,x_1)\sim\pi^*}\left[ \|v_{t,\theta}(x_{t,\eta^*}) - \dot{x}_{t,\eta^*}\|_{g_{\text{RBF}}(x_{t,\eta^*})}^2 \right]. \tag{13}$$

We note that the case of $g_{\text{LAND}}$ is dealt with similarly. Additionally, different choices of the joint distribution $q$, beyond Optimal Transport, can be adapted from CFM [Tong et al., 2023b] to MFM in eq. (7).

## 4.2 Understanding MFM through energies

In MFM we learn interpolants $x_t$ that are *optimal* according to eq. (6). Previous works have studied the "optimality" of interpolants, but have ignored the data manifold. Shaul et al. [2023] proposed to learn interpolants $x_t$ that minimize the **kinetic energy** $\mathcal{K}(\dot{x}_t) = \mathbb{E}_{t,(x_0,x_1)\sim q} \|\dot{x}_t\|^2$. However, $\mathcal{K}$ assigns each point in space the same cost, *independent of the data*, and leads to straight interpolants that may stray away from $\mathcal{D}$. Conversely, our objective in eq. (6) can equivalently be written as

$$\mathcal{L}_g(\eta) = \mathbb{E}_{(x_0,x_1)\sim q}\left[ \mathcal{E}_g(x_{t,\eta}) \right] \equiv \mathbb{E}_{t,(x_0,x_1)\sim q}[\|\dot{x}_{t,\eta}\|_{g(x_{t,\eta})}^2].$$

As a result, the geodesic loss $\mathcal{L}_g(\eta) \equiv \mathcal{K}_g(\dot{x}_{t,\eta})$ is *precisely* the kinetic energy of vector fields with respect to $g$ and hence accounts for the cost induced by the data. In fact, choosing $\mathbf{G}(x) = \mathbf{I}$, recovers

$\mathcal{K}$. We note that the parameterization $x_{t,\eta}$ in eq. (4) is more expressive than $x_t = a(t)x_1 + b(t)x_0$, which is studied in Albergo and Vanden-Eijnden [2022], Shaul et al. [2023]. Besides, $x_{t,\eta}$ not only depends on the endpoints $x_0, x_1$ but, implicitly, on all the data points $\mathcal{D}$ through metrics such as $g_{\text{RBF}}$.

**Data-dependent potentials**. Energies more general than the kinetic one $\mathcal{K}$ have been considered in Neklyudov et al. [2023a], Liu et al. [2024], Neklyudov et al. [2023b]. In particular, adapting GSBM [Liu et al., 2024] from the stochastic Schrödinger bridge setting to CFM, entails designing interpolants $x_t$ that minimize $\mathcal{U}(x_t, \dot{x}_t) = \mathcal{K}(\dot{x}_t) + V_t(x_t)$, with $V_t$ a potential enforcing additional constraints. However, GSBM does not prescribe a general recipe for $V_t$ and instead leaves to the modeler the task of constructing $V_t$, based on applications. Conversely, MFM relies on a Riemannian approach to propose an *explicit* objective, i.e. eq. (11), that holds irrespective of the task and can be learned *prior* to regressing the vector field $v_{t,\theta}$. In fact, eq. (11) can be rewritten as

$$\mathcal{L}_{g_{\text{RBF}}}(\eta) = \mathbb{E}_{t,(x_0,x_1)\sim q}\left[\|\dot{x}_{t,\eta}\|^2 + V_{t,\eta}(x_{t,\eta}, x_0, x_1)\right]. \tag{14}$$

where $V_{t,\eta}$ is parametric function depending on the boundary points $x_0, x_1$ (see Appendix §C.1 for an expression). In contrast to GSBM, MFM designs interpolants $x_{t,\eta}$ that jointly minimize $\mathcal{K}$ and a potential $V_{t,\eta}$ that is not fixed but also updated with the same parameters $\eta$ to bend paths towards the data.

## 5 Experiments

We test METRIC FLOW MATCHING on different tasks: artificial dynamic reconstruction and navigation through LiDAR surfaces §5.1; unpaired translation between classes in images §5.2; reconstruction of cell dynamics. Further results and experimental details can be found in Appendices §D, §E and §F.

**The model.** In all the experiments, we test the OT-MFM method detailed in §4.1. As argued in §4.1, for high-dimensional data, we leverage the RBF metric (10) and hence train with the objective (13). We refer to this model as OT-MFM$_{\text{RBF}}$. Conversely, for low-dimensional data we rely on the LAND metric (9), replacing $g_{\text{RBF}}$ with $g_{\text{LAND}}$ in eq. (11) and eq. (13). We denote this model as OT-MFM$_{\text{LAND}}$. *Both metrics are task-independent and do not require further manipulation from the modeler*. To assess the impact of metric learning in MFM even without using Optimal Transport for the coupling $q$, we tested MFM with independent coupling on the Arch task and the single cell datasets, denoted as I-MFM$_{\text{LAND}}$ and I-MFM$_{\text{RBF}}$.

**Baselines.** MFM generalizes CFM by learning interpolants that account for the geometry of the data. As such, in §5.1 and §5.2, we focus on validating that OT-MFM leads to more meaningful matching than its Euclidean counterpart OT-CFM [Tong et al., 2023b]. For single-cell experiments instead, we also compare with a variety of baselines, including models specific to the single-cell domain [Tong et al., 2020, Koshizuka and Sato, 2023], Schrödinger Bridge models [De Bortoli et al., 2021, Shi et al., 2023, Tong et al., 2023a], and ODE-flow methods [Tong et al., 2023b, Neklyudov et al., 2023b].

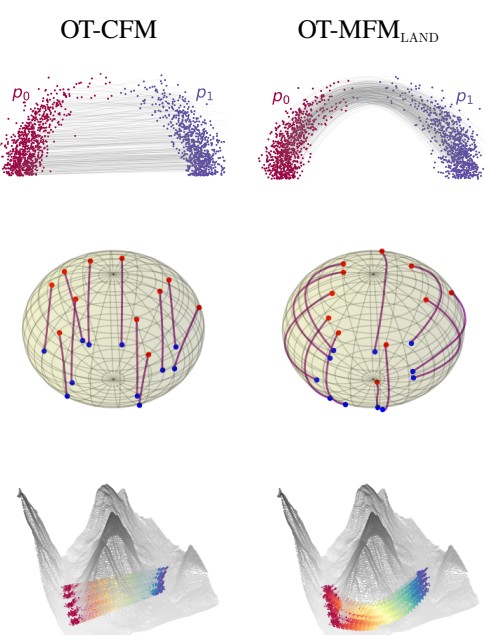

Figure 2: Interpolants for the Arch dataset (top row), Sphere dataset (middle row) and over LiDAR scans of Mt. Rainier (bottom row). In all cases, learning interpolants that minimize the LAND metric (9) leads to more meaningful matchings.

### 5.1 Synthetic experiments and LiDAR

Our first goal is to validate that MFM and specifically the family of interpolants $x_{t,\eta}$ in eq. (4), are expressive enough to model matching of distributions when the data induce a *nonlinear* geometry.

To begin with, we consider the Arch dataset in Tong et al. [2020], where we have access to the underlying paths. In Table 1 we report the Wasserstein distance (EMD) between the interpolation according to true dynamics and the marginal at time $1/2$ obtained using the Euclidean baseline OT-CFM and our Riemannian model OT-MFM$_{\text{LAND}}$. We see that OT-MFM$_{\text{LAND}}$ is able to faithfully reconstruct the dynamics by learning interpolants that minimize the geodesic cost induced by the metric in eq. (9). Conversely, straight interpolants fail to stay on the manifold generated by the dynamics (see Figure 2).

Table 1: Wasserstein distance between reconstructed marginal at time $1/2$ and ground-truth.

| Method | EMD ($\downarrow$) |
|---|---|
| I-CFM | $0.6120 \pm 0.014$ |
| OT-CFM | $0.6081 \pm 0.023$ |
| I-MFM$_{\text{LAND}}$ | $0.1210 \pm 0.020$ |
| OT-MFM$_{\text{LAND}}$ | $\mathbf{0.0813 \pm 0.009}$ |

To further showcase the ability of MFM to learn trajectories that stay close to an unknown underlying manifold without any added noise, we conduct an experiment on a 2D sphere embedded in $\mathbb{R}^3$. We consider source and target distributions that are Gaussians centered at the sphere's poles, with samples lying exactly on the sphere. MFM significantly improves over the Euclidean baseline, CFM, with samples at intermediate times being much closer to the underlying sphere than those from the Euclidean counterpart. This improvement is achieved *without explicitly parameterizing the lower-dimensional space*, relying solely on the LAND metric. Quantitative results, including EMD and the distance of the interpolating paths to the sphere, are reported in Appendix G.

Finally, to highlight the ability of MFM to generate meaningful matching, we also compare the interpolants of OT-CFM and OT-MFM$_{\text{LAND}}$ for navigations through surfaces scanned by LiDAR [Legg and Anderson, 2013]. From Figure 2 we find that straight paths result in unnatural trajectories, whereas OT-MFM manages to construct meaningful interpolations that better navigate the complex surface. While OT-MFM can be further enhanced using potentials specific to the task, similar to [Liu et al., 2024], here we focus on its ability to provide meaningful matching even with a task-agnostic metric.

## 5.2 Unpaired translation in latent space

To test the advantages of MFM for more meaningful generation, we consider the task of unpaired image translation between dogs and cats in AFHQ [Choi et al., 2020]. Specifically, we perform unpaired translation in the *latent space* of the Stable Diffusion v1 VAE [Rombach et al., 2022]. Figure 3 reports a qualitative comparison between OT-CFM and OT-MFM$_{\text{RBF}}$. Additionally, a quantitative comparison can be found in Table 2, where we measure both the quality of images via Fréchet Inception Distance (FID) [Heusel et al., 2017] and the perceptual similarity of generated cats to source dogs via Learned Perceptual Image Patch Similarity (LPIPS) [Zhang et al., 2018]. We see that OT-MFM improves upon the Euclidean baseline. Our results using MFM further highlight the role played by the nonlinear geometry associated with latent representations, a topic studied extensively §6.

| OT-CFM | | | | | OT-MFM$_{\text{RBF}}$ | | | | |
|---|---|---|---|---|---|---|---|---|---|
| t=0.00 | t=0.25 | t=0.50 | t=0.75 | t=1.00 | t=0.00 | t=0.25 | t=0.50 | t=0.75 | t=1.00 |

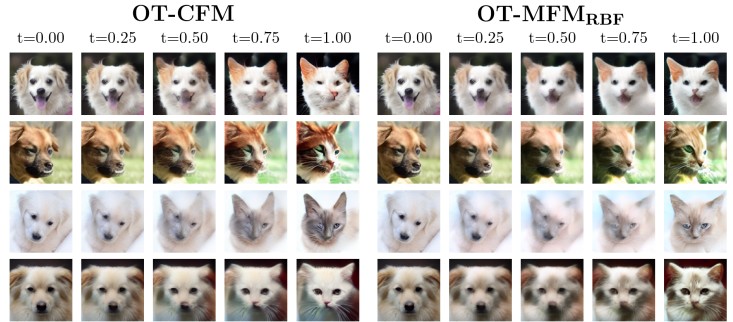

Figure 3: Qualitative comparison for image translation. By designing interpolants on the data manifold, OT-MFM$_{\text{RBF}}$ better preserves input features.

Table 2: FID ($\downarrow$) and LPIPS ($\downarrow$) values on AFHQ for OT-CFM and OT-MFM$_{\text{RBF}}$.

| Method | FID | LPIPS |
|---|---|---|
| OT-CFM | 41.42 | 0.512 |
| OT-MFM$_{\text{RBF}}$ | **37.87** | **0.502** |

## 5.3 Trajectory inference for single-cell data

We finally test MFM for reconstructing cell dynamics, a central problem in biomedical applications [Lähnemann et al., 2020], which holds great promise thanks to the advancements of single-cell RNA sequencing (scRNA-seq) [Macosko et al., 2015, Klein et al., 2015]. Since in scRNA-seq trajectories cannot be tracked, we only assume access to $K$ unpaired distributions describing cell populations at $K$ time points. We then apply the matching objective in eq. (7) between every consecutive time points, sharing parameters for both the vector field $v_{t,\theta}$ and the interpolants $x_{t,\eta}$—see eq. (20) for how to extend $x_{t,\eta}$ to multiple time-points. Following the setup of Schiebinger et al. [2019], Tong et al. [2020, 2023a] we perform leave-one-out interpolation, where we measure the Wasserstein-1 distance between the $k$-th left-out density and the one reconstructed after training on the remaining timepoints. We compare OT-MFM and baselines over Embryoid

Table 3: Wasserstein-1 distance averaged over left-out marginals for 100-dim PCA single-cell data for corresponding datasets. Results averaged over 5 runs.

| Method | Cite (100D) | Multi (100D) |
|---|---|---|
| SF$^2$ M-Geo | $44.498 \pm 0.416$ | $52.203 \pm 1.957$ |
| SF$^2$ M-Exact | $46.530 \pm 0.426$ | $52.888 \pm 1.986$ |
| OT-CFM | $45.393 \pm 0.416$ | $54.814 \pm 5.858$ |
| I-CFM | $48.276 \pm 3.281$ | $57.262 \pm 3.855$ |
| WLF-OT | $44.821 \pm 0.126$ | $55.416 \pm 6.097$ |
| WLF-UOT | $43.731 \pm 1.375$ | $54.222 \pm 5.827$ |
| WLF-SB | $46.131 \pm 0.083$ | $55.065 \pm 5.499$ |
| I-MFM$_{\text{RBF}}$ | $45.987 \pm 4.014$ | $54.197 \pm 1.408$ |
| OT-MFM$_{\text{RBF}}$ | $41.784 \pm 1.020$ | $50.906 \pm 4.627$ |

body (EB) [Moon et al., 2019], and CITE-seq (Cite) and Multiome (Multi) data from [Lance et al., 2022]. In Table 4 and Table 3 we consider the first 5 and 100 principal components of the data, respectively—results with 50 principal components can be found in Appendix D. We observe that OT-MFM significantly improves upon its Euclidean counterpart OT-CFM, which resonates with the manifold hypothesis for single-cell data [Moon et al., 2018]. In fact, OT-MFM surpasses all baselines, including those that add biases such as stochasticity (SF$^2$ Tong et al. [2023a]) or mass teleportation (WLF-UOT Neklyudov et al. [2023b]). OT-MFM instead relies on metrics such as LAND and RBF to favor interpolations that remain close to the data.

## 6 Related Work

**Geometry-aware generative models**. The manifold hypothesis [Bengio et al., 2013] has been studied in the context of manifold learning [Tenenbaum et al., 2000, Belkin and Niyogi, 2003] and metric learning [Xing et al., 2002, Weinberger and Saul, 2009, Hauberg et al., 2012]. Recently, this has also

Table 4: Wasserstein-1 distance ($\downarrow$) averaged over left-out marginals for 5-dim PCA representation of single-cell data for corresponding datasets. Results are averaged over 5 independent runs.

| Method | Cite | EB | Multi |
|---|---|---|---|
| Reg. CNF [Finlay et al., 2020] | — | $0.825 \pm 0.429$ | — |
| TrajectoryNet [Tong et al., 2020] | — | $0.848$ | — |
| NLSB [Koshizuka and Sato, 2023] | — | $0.970$ | — |
| DSBM [Shi et al., 2023] | $1.705 \pm 0.160$ | $1.775 \pm 0.429$ | $1.873 \pm 0.631$ |
| DSB [De Bortoli et al., 2021] | $0.953 \pm 0.140$ | $0.862 \pm 0.023$ | $1.079 \pm 0.117$ |
| SF$^2$ M-Sink [Tong et al., 2023a] | $1.054 \pm 0.087$ | $1.198 \pm 0.342$ | $1.098 \pm 0.308$ |
| SF$^2$ M-Geo [Tong et al., 2023a] | $1.017 \pm 0.104$ | $0.879 \pm 0.148$ | $1.255 \pm 0.179$ |
| SF$^2$ M-Exact [Tong et al., 2023a] | $0.920 \pm 0.049$ | $0.793 \pm 0.066$ | $0.933 \pm 0.054$ |
| OT-CFM [Tong et al., 2023b] | $0.882 \pm 0.058$ | $0.790 \pm 0.068$ | $0.937 \pm 0.054$ |
| I-CFM [Tong et al., 2023b] | $0.965 \pm 0.111$ | $0.872 \pm 0.087$ | $1.085 \pm 0.099$ |
| SB-CFM [Tong et al., 2023b] | $1.067 \pm 0.107$ | $1.221 \pm 0.380$ | $1.129 \pm 0.363$ |
| WLF-UOT [Neklyudov et al., 2023b] | $0.733 \pm 0.063$ | $0.738 \pm 0.014$ | $0.911 \pm 0.147$ |
| WLF-SB [Neklyudov et al., 2023b] | $0.797 \pm 0.022$ | $0.746 \pm 0.016$ | $0.950 \pm 0.205$ |
| WLF-OT [Neklyudov et al., 2023b] | $0.802 \pm 0.029$ | $0.742 \pm 0.012$ | $0.949 \pm 0.211$ |
| I-MFM$_{\text{LAND}}$ | $0.916 \pm 0.124$ | $0.822 \pm 0.042$ | $1.053 \pm 0.095$ |
| OT-MFM$_{\text{LAND}}$ | $0.724 \pm 0.070$ | $0.713 \pm 0.039$ | $0.890 \pm 0.123$ |

been analyzed in relation to generative models for obtaining meaningful interpolations [Arvanitidis et al., 2017, 2021, Chadebec and Allassonnière, 2022], diagnosing model instability [Cornish et al., 2020, Loaiza-Ganem et al., 2022, 2024], assessing the ability to perform dimensionality reduction [Stanczuk et al., 2022, Pidstrigach, 2022] and for the improved learning and representation of curved, low-dimensional data manifolds [Dupont et al., 2019, Schonsheck et al., 2019, Horvat and Pfister, 2021, Yonghyeon et al., 2021, De Bortoli, 2022, Jang et al., 2022, Lee et al., 2022, Lee and Park, 2023, Nazari et al., 2023]. Closely related is the extension of generative models to settings where the ambient space itself is a Riemannian manifold [Mathieu and Nickel, 2020, Lou et al., 2020, Falorsi, 2021, De Bortoli et al., 2022, Huang et al., 2022, Rozen et al., 2021, Ben-Hamu et al., 2022, Chen and Lipman, 2023, Jo and Hwang, 2023]. Particularly, Maoutsa [2023] utilized a data-dependent geodesic solver [Arvanitidis et al., 2019] to refine drift estimation in stochastic differential equations. Several studies extended beyond the standard linear matching process to meet specific task requirements, but have not accounted for the geometry that the data naturally forms [Liu et al., 2024, Bartosh et al., 2024, Neklyudov et al., 2023b].

**Trajectory inference**. Reconstructing dynamics from cross-sectional distributions [Hashimoto et al., 2016, Lavenant et al., 2021] is an important problem within the natural sciences, especially in the context of single-cell analysis [Macosko et al., 2015, Moon et al., 2018, Schiebinger et al., 2019]. Recently, diffusion and Continuous Normalizing Flow (CNFs) based methods have been proposed [Tong et al., 2020, Bunne et al., 2022, 2023, Huguet et al., 2022, Koshizuka and Sato, 2023] but require simulations, whereas Tong et al. [2023a], Neklyudov et al. [2023a], Palma et al. [2023] allow for simulation-free training. In particular, Huguet et al. [2022], Palma et al. [2023] regularize CNFs in a latent space to enforce that straight paths correspond to interpolations on the original data manifold. Finally, Scarvelis and Solomon [2023] propose a solution to the trajectory inference problem for cellular data, which depends on a regularization of vector fields with respect to a learned metric similar to eq. (6). Crucially though, their framework requires simulations in training and is not immediately extended to more general matching objectives and applications as for MFM. In this regard, we observe that one can also adopt the metric-learning scheme of Scarvelis and Solomon [2023] in MFM, replacing $g_{\text{LAND}}$ and $g_{\text{RBF}}$ introduced in §4.

# 7 Conclusions and Limitations

We have presented METRIC FLOW MATCHING, a simulation-free framework that generalizes Conditional Flow Matching to design probability paths whose support lies on the data manifold. In MFM, this is achieved via interpolants that minimize the geodesic cost of a data-dependent Riemannian metric. We have empirically shown that instances of MFM using prescribed task-agnostic metrics, surpass Euclidean baselines, with emphasis on single-cell dynamics reconstruction. While the universality of the metrics proposed in §4 is a benefit, we have not investigated how to further encode biases into the metric that are specific to the downstream task—a topic reserved for future work. Additionally, the principle of learning interpolants that minimize a geodesic cost can also be adapted to score-based generative models such as diffusion models, beyond CFM. When relying on the OT coupling, standard limitations of using OT for high-dimensional problems with large datasets may arise. Lastly, our approach requires the data to be embedded in Euclidean space for the interpolants to be defined; it is an interesting direction to explore how one can learn interpolants that minimize a data-dependent metric even when the ambient space itself is not Euclidean.

# Acknowledgements

KK is supported by the EPSRC CDT in Health Data Science (EP/S02428X/1). PP and LZ are supported by the EPSRC CDT in Modern Statistics and Statistical Machine Learning (EP/S023151/1) and acknowledge helpful discussions with Yee Whye Teh. AJB is partially supported by an NSERC Post-doc fellowship and an EPSRC Turing AI World-Leading Research Fellowship. FDG is supported by an EPSRC Turing AI World-Leading Research Fellowship. MB is partially supported by EPSRC Turing AI World-Leading Research Fellowship No. EP/X040062/1 and EPSRC AI Hub on Mathematical Foundations of Intelligence: An "Erlangen Programme" for AI No. EP/Y028872/1

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

## Outline of Appendix

In Appendix A we give a brief overview of relevant notions from differential and Riemannian geometry. Appendix B provides more details for Section 3 including the formal statement and proof of Proposition 1. We also derive how to rigorously extend flow matching to the Riemannian manifold induced by a data-dependent metric. We report a pseudocode for MFM in Algorithm 2. Appendix C provides more details regarding the data-dependent Riemannian metrics we use, and relevant training procedures. In Appendix D we supply more details regarding the various experiments. Appendix E, Appendix F, and Appendix H contain supplementary figures and tables for single-cell reconstruction, unpaired translation, and LiDAR tasks, respectively. Finally, in Appendix G, we outline the quantitative evaluation results for MFM on the sphere experiment.

**All exact hyperparameters used and reproducible code can be found at** https://github.com/kksniak/metric-flow-matching.git.

## A  Primer on Riemannian Geometry

Riemannian geometry is the study of smooth manifolds $\mathcal{M}$ equipped with a (Riemannian) metric $g$. Intuitively, this corresponds to spaces that can be considered locally Euclidean, and which allow for a consistent notion of measuring distances, angles, curvature, shortest paths, etc. on abstract geometric objects. We provide a primer on the relevant concepts of Riemannian geometry for our work, covering smooth manifolds and their tangent spaces, Riemannian metrics and geodesics, and integration on Riemannian manifolds. For a comprehensive introduction, see Lee [2012].

**Smooth manifolds**. Formally, we say a topological space[2] $\mathcal{M}$ is a $d$-dimensional smooth manifold if we have a collection of charts $(U_i, \varphi_i)$ where $U_i$ are open subsets of $\mathcal{M}$ and $\cup_i U_i = \mathcal{M}$, $\varphi_i : U_i \to \mathbb{R}^d$ are homeomorphisms onto their image, and when $U_i \cap U_j \neq \emptyset$, the transition maps $\varphi_j \circ \varphi_i^{-1} : \varphi_i(U_i \cap U_j) \to \varphi_j(U_i \cap U_j)$ are diffeomorphisms. These charts allow us to represent quantities on $\mathcal{M}$ through the local coordinates obtained by mapping back to Euclidean space via $\varphi_i$, as well as allowing us to define smooth maps between manifolds.

In particular, we define a smooth path in $\mathcal{M}$ passing through $x \in \mathcal{M}$ as a function $\gamma : (-\epsilon, \epsilon) \to \mathcal{M}$, for some $\epsilon > 0$ and $\gamma(0) = x$, such that the local coordinate representation $\varphi_i \circ \gamma$ of $\gamma$ is smooth in the standard sense (for any suitable chart $(U_i, \varphi_i)$). The derivatives $\dot{\gamma}(0)$ of smooth paths $\gamma$ passing through $x \in \mathcal{M}$ form a $d$-dimensional vector space called the tangent space $T_x\mathcal{M}$ at $x$. We can represent $T_x\mathcal{M}$ in local coordinates with $\varphi_i$ by the identification of $\dot{\gamma}(0) \in T_x\mathcal{M}$ to $(a_1'(0), \ldots, a_d'(0))^\top \in \mathbb{R}^d$ where $(a_1(t), \ldots, a_d(t))^\top = \varphi_i \circ \gamma(t)$.

**Riemannian structure**. To introduce geometric notions of length and distances to smooth manifolds, we define a Riemannian metric[3] $g$ on $\mathcal{M}$ as a map providing a smooth assignment of points $x \in \mathcal{M}$ on the manifold to a positive definite inner product $\langle \cdot, \cdot \rangle_{g(x)}$ defined on the corresponding tangent space $T_x\mathcal{M}$. In local coordinates about $x \in \mathcal{M}$ given by a suitable chart $(U_i, \varphi_i)$, we have the local representation $\langle v, w \rangle_{g(x)} = v^\top \mathbf{G}(x)w$ where $\mathbf{G}(x) \in \mathbb{R}^{d \times d}$ is a positive definite matrix (which implicitly depends on the choice of chart). We call the pair $(\mathcal{M}, g)$ a Riemannian manifold.

Through the Riemannian metric, we can now define the norm of tangent vectors by $\|v\|_{g(x)} = \langle v, v \rangle_{g(x)}^{1/2}$ for $v \in T_x\mathcal{M}$, as well as the length of a smooth path $\gamma : [0, 1] \to \mathcal{M}$ by

$$\text{Len}(\gamma) = \int_0^1 \|\dot{\gamma}(t)\|_{g(\gamma(t))} dt. \tag{15}$$

We can then define a geodesic $\gamma^*$ between $x_0$ and $x_1$ in $\mathcal{M}$ as the shortest possible path between the two points - i.e.

$$\gamma^* = \arg\min_\gamma \int_0^1 \|\dot{\gamma}(t)\|_{g(\gamma(t))} dt, \quad \gamma(0) = x_0, \ \gamma(1) = x_1. \tag{16}$$

---

[2]With the technical assumptions that $\mathcal{M}$ is second-countable and Hausdorff.

[3]Formally, this is a choice of a smooth symmetric covariant 2-tensor field on $\mathcal{M}$ which is positive definite at each point.

We assume that all Riemannian manifolds being considered are (geodesically) complete, meaning that geodesics can be extended indefinitely. In particular, for any pair of points $x_0, x_1$, there exists a unique geodesic $\gamma_t^*$ starting at $x_0$ and ending at $x_1$.

**Integration on Riemannian manifolds**. To introduce integration on Riemannian manifolds, we use the fact that under the technical assumption that $\mathcal{M}$ is orientable, the Riemannian manifold $(\mathcal{M}, g)$ has a canonical volume form $d\mathrm{vol}$. This can be used to define a measure on $\mathcal{M}$, where in local coordinates, we have that $d\mathrm{vol}(x) = \sqrt{|\mathbf{G}(x)|}dx$ where $dx$ denotes the Lebesgue measure in $\mathbb{R}^d$ and $|\cdot|$ denotes the determinant. Hence, for a chart $(U_i, \varphi_i)$ and a continuous function $f : \mathcal{M} \to \mathbb{R}$ which is compactly supported in $U_i$, we define the integral

$$\int_{\mathcal{M}} f d\mathrm{vol} = \int_{\varphi_i(U_i)} f \circ \varphi_i^{-1}(x) \sqrt{|\mathbf{G}(x)|} dx. \tag{17}$$

This definition can be extended to more general functions through the use of partitions of unity and the Riesz–Markov–Kakutani representation theorem.

**The Riemannian geometry of Metric Flow Matching**. Finally, we note that in our work, we take our smooth manifold to be $\mathcal{M} = \mathbb{R}^d$ with the trivial chart $U_i = \mathbb{R}^d, \varphi_i = \mathrm{id}$. This means we can work in the usual Euclidean coordinates instead of requiring charts and local coordinates, simplifying our framework. For example, we can define $g$ through $\mathbf{G}$ directly in Definition 1 without the need to check consistency across the choice of local coordinates, and we have the trivial identification of $T_x\mathcal{M}$ to $\mathbb{R}^d$. In addition, for a continuous $f : \mathcal{M} \to \mathbb{R}$, we have the simplified definition of the Riemannian integral (when the integral exists) as

$$\int_{\mathcal{M}} f d\mathrm{vol} = \int_{\mathbb{R}^d} f(x) \sqrt{|\mathbf{G}(x)|} dx. \tag{18}$$

We use this to define probability densities on $(\mathcal{M}, g)$ to extend CFM to our setting in §B.1.

# B    Additional Details for Section 3

To begin with, we provide a formal statement covering Proposition 1 and report a proof below. We introduce some notation. We write $\gamma_t \subset U$ when the image set of $\gamma : [0, 1] \to \mathbb{R}^d$ is contained in $U$, i.e. $\gamma_t \in U$ for each $t \in [0, 1]$. Moreover, we let

$$B_r(\mathcal{D}) := \{y \in \mathbb{R}^d : \exists x_i \in \mathcal{D} \text{ s.t. } \|y - x_i\| \le r\},$$

denote the set of points in $\mathbb{R}^d$ whose distance from the dataset $\mathcal{D}$ is at most $r > 0$.

**Theorem B.1** (Formal statement of Proposition 1). *Consider a closed dataset $\mathcal{D} \subset \mathbb{R}^d$ (e.g. finite). Assume that for each $(x_0, x_1) \sim q$, there exists at least a path $\gamma_t$ connecting $x_0$ to $x_1$ whose length is at most $\Gamma$ and such that $\gamma_t \subset B_\delta(\mathcal{D})$, with $\delta > 0$. Let $\kappa > 0, \rho > \delta$ and $g$ be any data-dependent metric satisfying: (i) $v^\top \mathbf{G}(x; \mathcal{D})v \ge \kappa\|v\|^2$ for each $x \in \mathbb{R}^d \setminus B_\rho(\mathcal{D})$ and $v \in \mathbb{R}^d$; (ii) $\|\mathbf{G}(x; \mathcal{D})\| \le \kappa(\rho/\Gamma)^2$ for each $x \in B_\delta(\mathcal{D})$. Then for any $(x_0, x_1) \sim q$, the geodesic $\gamma_t^*$ of $g$ connecting $x_0$ to $x_1$ satisfies $\gamma_t^* \subset B_{2\rho}(\mathcal{D})$.*

*Proof of Theorem B.1.* We argue by contradiction and assume that there exist $x \in \mathbb{R}^d \setminus B_{2\rho}(\mathcal{D})$ and $(x_0, x_1) \sim q$ such that the geodesic $\gamma_t^*$ of $g$ connecting $x_0$ to $x_1$ passes through $x$, i.e. there is a time $t_0 \in (0, 1)$ such that $\gamma_{t_0}^* = x$.

By assumption, there exists a path $\gamma_t$ that connects $x_0$ to $x_1$ and stays within $B_\delta(\mathcal{D})$, which means that $x_0 = \gamma_0 = \gamma_0^* \in B_\delta(\mathcal{D})$. Since $\mathcal{D}$ is closed, the function $x \mapsto d_{\mathrm{E}}(x, \mathcal{D}) := \inf_{y \in \mathcal{D}} \|x - y\|$, i.e. the Euclidean distance of $x$ from the dataset $\mathcal{D}$, is continuous. In particular, $t \mapsto d_{\mathrm{E},\mathcal{D}}(t) := d_{\mathrm{E}}(\gamma_t^*, \mathcal{D})$ is also continuous, due to the geodesic being a smooth function in the interval $[0, 1]$. From the continuity of $d_{\mathrm{E},\mathcal{D}}$ and the fact that $d_{\mathrm{E},\mathcal{D}}(0) \le \delta$ and $d_{\mathrm{E},\mathcal{D}}(t_0) > 2\rho$, it follows that there must be a time $0 < t' < t_0$ such that $d_{\mathrm{E},\mathcal{D}}(t) \ge \rho$ for all $t \in (t', t_0]$ and $d_{\mathrm{E},\mathcal{D}}(t') = \rho$.

If we unpack the definition of $d_{\mathrm{E},\mathcal{D}}$, we have just shown that $\gamma_t^* \in \mathbb{R}^d \setminus B_\rho(\mathcal{D})$ for all $t \in (t', t_0]$. Accordingly, we can estimate the length of $\gamma_t^*$ with respect to the Riemannian metric $g$ as

$$\begin{aligned}
\mathrm{Len}_g(\gamma_t^*) = \int_0^1 \|\dot\gamma_t^*\|_{g(\gamma_t^*)} \, dt &> \int_{t'}^{t_0} \|\dot\gamma_t^*\|_{g(\gamma_t^*)} \, dt \\
&= \int_{t'}^{t_0} \sqrt{(\dot\gamma_t^*)^\top \mathbf{G}(\gamma_t^*; \mathcal{D})\dot\gamma_t^*} \, dt \ge \sqrt{\kappa} \int_{t'}^{t_0} \|\dot\gamma_t^*\| \, dt \ge \sqrt{\kappa}\|\gamma_{t'}^* - \gamma_{t_0}^*\|,
\end{aligned}$$

where in the second-to-last inequality we have used the assumption (i) on the metric having minimal eigenvalue $\sqrt{\kappa}$ in $\mathbb{R}^d \setminus B_\rho(\mathcal{D})$, while the final inequality simply follows from the Euclidean length of any curve between the points $\gamma_{t'}^*$ and $\gamma_{t_0}^*$ being larger than their Euclidean distance.

We claim that $\|\gamma_{t'}^* - \gamma_{t_0}^*\| \geq \rho$. To validate the latter point, take $\epsilon > 0$. It follows that there exists $x_i \in \mathcal{D}$ such that $\|x_i - \gamma_{t'}^*\| \leq \rho + \epsilon$, because $d_{\mathrm{E},\mathcal{D}}(t') = \rho$, i.e. $\gamma_{t'}^* \in B_\rho(\mathcal{D})$. If $\|\gamma_{t'}^* - \gamma_{t_0}^*\| < \rho$, then we could apply the triangle inequality and derive that

$$\|\gamma_{t_0}^* - x_i\| \leq \|\gamma_{t_0}^* - \gamma_{t'}^*\| + \|\gamma_{t'}^* - x_i\| < \rho + \rho + \epsilon = 2\rho + \epsilon.$$

Since $\epsilon > 0$ was arbitrary, it would follow that $\gamma_{t_0}^* \equiv x \in B_{2\rho}(\mathcal{D})$, which is a contradiction to our starting point. Therefore, $\|\gamma_{t'}^* - \gamma_{t_0}^*\| \geq \rho$, and hence

$$\mathrm{Len}_g(\gamma_t^*) > \sqrt{\kappa}\rho. \tag{19}$$

On the other hand, the assumptions guarantee the existence of another path $\gamma_t$ connecting $x_0$ to $x_1$ whose image is always contained in $B_\delta(\mathcal{D})$. It follows that

$$\mathrm{Len}_g(\gamma_t) = \int_0^1 \|\dot{\gamma}_t\|_{g(\gamma_t)}\, dt = \int_0^1 \sqrt{(\dot{\gamma}_t^*)^\top \mathbf{G}(\gamma_t^*; \mathcal{D})\dot{\gamma}_t^*}\, dt$$

$$\leq \sqrt{\kappa}\frac{\rho}{\Gamma} \int_0^1 \|\dot{\gamma}_t\|\, dt = \sqrt{\kappa}\frac{\rho}{\Gamma}\Gamma = \sqrt{\kappa}\rho,$$

where we have used the assumption (ii) on $g$ and that the length of $\gamma_t$ is at most $\Gamma$. We can then combine the last inequality and eq. (19), and conclude that

$$\mathrm{Len}_g(\gamma_t) \leq \sqrt{\kappa}\rho < \mathrm{Len}_g(\gamma_t^*),$$

which means that we have found a path connecting $x_0$ to $x_1$, whose length with respect to the metric $g$ is shorter than the one of the geodesic $\gamma_t^*$ from $x_0$ to $x_1$. This is a contradiction and concludes the proof. $\qquad\square$

## B.1 Extending CFM to the manifold $(\mathbb{R}^d, g)$

In this subsection, we demonstrate how to generalize the CFM objective in eq. (1) to the Riemannian manifold $(\mathbb{R}^d, g)$, defined by a data-dependent metric $g$ as per Definition 1.

**Densities on the manifold**. To begin with, we extend the densities $p_0, p_1$ and the joint density $q$ to valid densities on the manifold $(\mathbb{R}^d, g)$. First, we recall that the Riemannian volume form induced by $g$ can be written in coordinates as

$$d\mathrm{vol}(x) = \sqrt{|\mathbf{G}(x; \mathcal{D})|}dx,$$

where $|\cdot|$ denotes the determinant of the matrix $\mathbf{G}(x; \mathcal{D})$, while $dx$ is the standard Lebesgue measure on $\mathbb{R}^d$. Accordingly, given a density $p \in \mathbb{P}(\mathbb{R}^d)$, we can derive the associated density $\hat{p} \in \mathbb{P}(\mathbb{R}^d, g)$ by rescaling:

$$\hat{p}(x) := \frac{p(x)}{\sqrt{|\mathbf{G}(x; \mathcal{D})|}},$$

which guarantees that $\hat{p}$ is continuous, positive, and

$$\int_{\mathbb{R}^d} \hat{p}(x)d\mathrm{vol}(x) = \int_{\mathbb{R}^d} p(x)dx = 1.$$

This in particular applies to the marginals $p_0, p_1$ given by our problem. A similar argument applies to the joint density $q$ whose marginals are $p_0$ and $p_1$, respectively. In fact, we can now define a density $\hat{q}$ on the product manifold, i.e. $\hat{q} \in \mathbb{P}(\mathbb{R}^d \times \mathbb{R}^d, g \times g)$, by simply taking

$$\hat{q}(x_0, x_1) := \frac{q(x_0, x_1)}{\sqrt{|\mathbf{G}(x_0; \mathcal{D}) \cdot \mathbf{G}(x_1; \mathcal{D})|}},$$

where the denominator is exactly the pointwise volume form induced by the product metric $g \times g$. Therefore, the MFM objective in (7) is interpreted as

$$\mathcal{L}_{\mathrm{MFM}}(\theta) = \mathbb{E}_{t,(x_0,x_1)\sim\hat{q}}\left[\|v_{t,\theta}(x_{t,\eta^*}) - \dot{x}_{t,\eta^*}\|^2_{g(x_{t,\eta^*})}\right]$$

$$= \mathbb{E}_t \int_{\mathbb{R}^d \times \mathbb{R}^d} \|v_{t,\theta}(x_{t,\eta^*}) - \dot{x}_{t,\eta^*}\|^2_{g(x_{t,\eta^*})} \hat{q}(x_0,x_1)\sqrt{|\mathbf{G}(x_0;\mathcal{D})\cdot\mathbf{G}(x_1;\mathcal{D})|}dx_0dx_1$$

$$= \mathbb{E}_t \int_{\mathbb{R}^d \times \mathbb{R}^d} \|v_{t,\theta}(x_{t,\eta^*}) - \dot{x}_{t,\eta^*}\|^2_{g(x_{t,\eta^*})} q(x_0,x_1)dx_0dx_1$$

$$= \mathbb{E}_{t,(x_0,x_1)\sim q}\left[\|v_{t,\theta}(x_{t,\eta^*}) - \dot{x}_{t,\eta^*}\|^2_{g(x_{t,\eta^*})}\right],$$

which is exactly eq. (7). We highlight that for this reason, we slightly abuse notation, and write an expectation with respect to $\hat{q}$, regarded as a density on the manifold $(\mathbb{R}^d, g)$, the same as an expectation over the density $q$ with respect to Lebesgue measure.

**The matching objective**. We also emphasize that, similarly to Riemannian Flow Matching [Chen and Lipman, 2023], we normalize the regression objective by $\mathbf{G}^{-1/2}(x;\mathcal{D})$ to avoid the need for initialization schemes that are specific to the metric. In fact, introducing a high-dimensional, non-trivial norm $\|\cdot\|_g$ in the CFM objective could introduce instabilities in the optimization of CFM. As such, the objective in $\mathcal{L}_{\mathrm{MFM}}$ effectively reduces to $\mathcal{L}_{\mathrm{CFM}}$ with interpolants $x_{t,\eta^*}$ minimizing the geodesic loss $\mathcal{L}_g$. Equivalently, METRIC FLOW MATCHING consists of a 2-step procedure: we first learn interpolants that approximate the geodesics of a data-dependent metric (Algorithm 1), and then we regress the vector field $v_\theta$ using the interpolants from the first stage as per the standard CFM objective. A pseudocode for the MFM-pipeline is reported in Algorithm 2.

**Inference**. We finally note that, in principle, the differential equation generated by $v_\theta$ is defined over the tangent bundle of $(\mathbb{R}^d, g)$ and hence solving it through Euler discretization, would require adopting the exponential map associated with the metric $g$ to project the tangent vector to the manifold. However, since the underlying space is still $\mathbb{R}^d$, the Euclidean Euler-step integration provides a first-order approximation of the Riemannian exponential associated with $g$ (see for example Monera et al. [2014]). Accordingly, at inference, we can simply rely on the canonical Euler-discrete integration to approximate the exponential map associated with the data-dependent metric $g$, provided that the step size is small.

---

**Algorithm 2** Pseudocode for METRIC FLOW MATCHING

**Require:** coupling $q$, initialized network $v_{t,\theta}$, **trained** network $\varphi_{t,\eta}$, data-dependent metric $g(\cdot)$
1: **while** Training **do**
2:     Sample $(x_0, x_1) \sim q$ and $t \sim \mathcal{U}(0,1)$
3:     $x_{t,\eta} \leftarrow (1-t)x_0 + tx_1 + t(1-t)\varphi_{t,\eta}(x_0,x_1)$                        ▷ *eq. (4)*
4:     $\dot{x}_{t,\eta} \leftarrow x_1 - x_0 + t(1-t)\dot{\varphi}_{t,\eta}(x_0,x_1) + (1-2t)\varphi_{t,\eta}(x_0,x_1)$
5:     $\ell(\theta) \leftarrow \|v_{t,\theta}(x_{t,\eta}) - \dot{x}_{t,\eta}\|^2_{g(x_{t,\eta})}$        ▷ *Estimate of objective $\mathcal{L}_{\mathrm{MFM}}(\theta)$ from eq. (7)*
6:     Update $\theta$ using gradient $\nabla_\theta \ell(\theta)$
    **return** vector field $v_{t,\theta}$

---

### B.1.1 Support of $p_t$

We conclude this section by justifying our claims about the relation of the support of the probability path $p_t$ and the interpolants. Consider a family of conditional paths $p_t(x|x_0,x_1)$ such that $p_t(x|x_0,x_1) \approx \delta(x - x_t)$, where $x_t$ are the interpolants connecting $x_0$ to $x_1$, and $\delta$ is the Dirac distribution. Assume that $x_t$ are straight interpolants, i.e. for any $(x_0,x_1) \sim q$, we define $x_t = tx_1 + (1-t)x_0$. Let us now consider $y \in \mathbb{R}^d$ such that there is no $(x_0,x_1) \sim q : y = tx_1 + (1-t)x_0$. Accordingly:

$$p_t(y) := \int_{\mathbb{R}^d \times \mathbb{R}^d} p_t(y|x_0,x_1)q(x_0,x_1)dx_0dx_1 = \int_{\mathbb{R}^d \times \mathbb{R}^d} \delta(y - x_t)q(x_0,x_1)dx_0dx_1 = 0,$$

since there is no $(x_0,x_1)$ in the support of $q$, such that $y = x_t$. We have then shown that

$$\mathrm{supp}(p_t) \subset \{y \in \mathbb{R}^d : \exists (x_0,x_1) \sim q : y = tx_1 + (1-t)x_0\},$$

which can be limiting for tasks where the data induce a nonlinear geometry, as validated in Section 5.

## C   Additional Details on the Riemannian Metrics used

In this Section, we provide further details on the diagonal metrics introduced in Section 4, which we adopt in METRIC FLOW MATCHING. In particular, we comment on important differences between the LAND metric and the RBF metric. We finally derive an explicit connection between MFM and recent methods that learn interpolant minimizing generalized energies.

**Learning the RBF metric** $g_{\text{RBF}}$. For the RBF metric in (10), we follow the metric design of Arvanitidis et al. [2021] and find the centroids by performing k-means clustering. Similarly, we define the bandwidth $\lambda_k$ associated with cluster $C_k$ as:

$$\lambda_k = \frac{1}{2}\left(\frac{\kappa}{|C_k|}\sum_{x\in C_k}\|x - \hat{x}_k\|^2\right)^{-2},$$

where $\kappa$ is a tunable hyperparameter. We note that the bandwidth is chosen to assign smaller decay in (10) to the centroids of clusters that have high spread to better enable attraction of trajectories. Finally, the weights $\omega_{\alpha,k}$ are determined by training the loss function,

$$\mathcal{L}_{\text{RBF}}(\{\omega_{\alpha,k}\}) = \sum_{x_i\in\mathcal{D}}\left(1 - \tilde{h}_\alpha(x_i)\right)^2 = \sum_{x_i\in\mathcal{D}}\left(1 - \sum_{k=1}^K \omega_{\alpha,k}\exp\left(-\frac{\lambda_{\alpha,k}}{2}\|x_i - \hat{x}_k\|^2\right)\right)^2.$$

Two important comments are in order. First, by learning the RBF metric, the framework MFM effectively entails jointly learning a data-dependent Riemannian metric *and* a suitable matching objective along approximate geodesics $x_{t,\eta}$. Second, we note that more general training objectives can be adopted for $\tilde{h}$, to enforce specific properties for the metric that are task-aware. We reserve the exploration of this topic for future work.

**LAND vs RBF metrics**. While similar in spirit, since they both assign a lower cost to regions of space close to the support of the data points $\mathcal{D}$, the LAND metric [Arvanitidis et al., 2016] and the RBF metric [Arvanitidis et al., 2021] differ in two fundamental aspects. Crucially, RBF is learned based on the data points, requiring less tuning than LAND, which is a key advantage in high-dimensions and the main motivation for why we resort to RBF for experiments on images and single-cell data with more than 50 principal components. Additionally, the RBF metric assigns similar cost to regions with data and is, in principle, more robust than the LAND metric to variations in the concentration of samples in $\mathcal{D}$. While this is neither a benefit nor a downside in general, we observe that if a metric such as LAND consistently assigns much lower cost to regions of space with higher data concentration, then the geodesic objective in eq. (6) could always bias to learn interpolants $x_{t,\eta}$ moving through these regions independent of the starting point—this is a consequence of the fact that we never compute the actual length of the paths to avoid simulations. Note though that this has not been observed as an issue in practice whenever we adopted the LAND metric for low-dimensional data.

### C.1   Connection between Riemannian approach and data potentials

We detail how the geometric loss (6) used to learn interpolants $x_{t,\eta}$ can be recast as a generalization of the GSBM framework in Liu et al. [2024]. In general, for any data-dependent metric $g$, we can indeed rewrite $\mathcal{L}_g(\eta)$ in eq. (6) as:

$$\begin{aligned}\mathcal{L}_g(\eta) &= \mathbb{E}_{t,(x_0,x_1)\sim q}[\langle \dot{x}_{t,\eta}, \mathbf{G}(x_{t,\eta};\mathcal{D})\dot{x}_{t,\eta}\rangle]\\ &= \mathbb{E}_{t,(x_0,x_1)\sim q}\left[\|\dot{x}_{t,\eta}\|^2 + \langle \dot{x}_{t,\eta}, (\mathbf{G}(x_{t,\eta};\mathcal{D}) - \mathbf{I})\dot{x}_{t,\eta}\rangle\right]\\ &= \mathbb{E}_{t,(x_0,x_1)\sim q}\left[\|\dot{x}_{t,\eta}\|^2 + V_{t,\eta}(x_{t,\eta},x_0,x_1)\right],\end{aligned}$$

where the parametric potential $V_{t,\eta}$ has the form

$$\begin{aligned}V_{t,\eta}(x_{t,\eta},x_0,x_1) &= \langle \dot{x}_{t,\eta}, (\mathbf{G}(x_{t,\eta};\mathcal{D}) - \mathbf{I})\dot{x}_{t,\eta}\rangle\\ \dot{x}_{t,\eta} &= x_1 - x_0 + t(1-t)\dot{\varphi}_{t,\eta}(x_0,x_1) + (1-2t)\varphi_{t,\eta}(x_0,x_1),\end{aligned}$$

where we have used the parameterization of $x_{t,\eta}$ in eq. (4). By replacing $\mathbf{G}(x;\mathcal{D})$ with the explicit expression given by the RBF metric in eq. (10), this provides a concrete formulation for eq. (14). We

also note that an equivalent perspective amounts to replacing the parametric potential with a *fixed* function $\mathcal{U}_t(x_{t,\eta}, \dot{x}_{t,\eta})$ that depends not only on $x_{t,\eta}$ as the potentials in Liu et al. [2024], but also on the velocities $\dot{x}_{t,\eta}$. Once again, we emphasize that our framework prescribes explicit parametric potentials $V_{t,\eta}$ via the diagonal Riemannian metrics in Section 3.1 and hence differs from Liu et al. [2024] which leave to the user the task of designing potentials based on applications.

# D    Experimental Details

We parameterize the models $\varphi_{t,\eta}(x_0, x_1)$ in eq. (4) and $v_{t,\theta}(x_t)$ in eq. (7) as neural networks, and train them **separately and sequentially**. We start with $\varphi_{t,\eta}(x_0, x_1)$, so that interpolants are learned prior to regressing $v_{t,\theta}(x_t)$ in the matching objective. Synthetic Arch, Sphere, LiDAR, and single-cell experiments were run on a single CPU. A single run of the LiDAR architecture and 5-dimensional single-cell experiments trains in under 10 minutes, while higher-dimensional single-cell experiments typically train in under an hour. The unpaired translation experiment on AFHQ was trained on a GPU cluster with NVIDIA A100 and V100 GPUs. Training time for AFHQ varies by GPU, ranging from 12 hours (A100) to 1 day (V100).

## D.1    Synthetic Arch, Sphere and LiDAR experiments

In the synthetic Arch, Sphere and LiDAR experiments, we parameterized both $\varphi_{t,\eta}(x_0, x_1)$ and $v_{t,\theta}(x_t)$ as 3-layer MLP networks with a width of 64 and SeLU activation. The networks were trained for up to 1000 epochs with early stopping (patience of 3 based on validation loss). We employed the Adam optimizer Kingma and Ba [2014] with a learning rate of 0.0001 for $\varphi_{t,\eta}(x_0, x_1)$ and the AdamW optimizer Loshchilov and Hutter [2017] with a learning rate of $10^{-3}$ and a weight decay of $10^{-5}$ for $v_{t,\theta}(x_t)$. We used a 90%/10% train/validation split. Training samples served as source and target distributions and for calculating the LAND metric, while validation samples were used for early stopping. We used $\sigma = 0.125$ and $\epsilon = 0.001$ in the LAND metric. During inference, we solved for $p_t$ using the Euler integrator for 100 steps.

**Synthetic Arch**    To generate the Arch dataset, we follow the experimental setup from [Tong et al., 2020]. We sampled 5000 points from two half Gaussians $N(0, \frac{1}{2\pi})$ and $N(1, \frac{1}{2\pi})$. The exact optimal transport interpolant at $t = \frac{1}{2}$ was used as the test distribution. We then embedded the points on a half circle of radius 1 with noise $N(0, 0.1)$ added to the radius. The Earth Mover's Distance between the sampled and test sets was averaged across five independent runs.

**Synthetic Sphere**    To generate the Sphere dataset, we sampled 5,000 points. We sampled the latitudes from two half Gaussians $N(0, \frac{1}{2\pi})$ and $N(1, \frac{1}{2\pi})$, and then scaled them by $\pi$. We sampled longitudes uniformly. The exact optimal transport interpolant at $t = \frac{1}{2}$ was used as the test distribution. We then embedded the points on a sphere of radius 1 without any noise. The Earth Mover's Distance between the sampled and test sets, and the mean distance of the middle points of trajectories to the sphere (both reported in Appendix G) were averaged across five independent runs.

**LiDAR**    For LiDAR, the data consists of point clouds within $[-5, 5]^3 \subset \mathbb{R}^3$, representing scans of Mt. Rainier [Legg and Anderson, 2013]. The source and target distributions are generated as Gaussian Mixture Models and projected onto the LiDAR manifold as in Liu et al. [2024]. We then standardized all the LiDAR, source, and target points.

## D.2    Unpaired translation in latent space

In the unpaired translation experiments, we utilized the U-Net architecture setup from Dhariwal and Nichol [2021] for both $\varphi_{t,\eta}(x_0, x_1)$ and $v_{t,\theta}(x_t)$. The exact hyperparameters are reported in Table 5. We used the Adam optimizer Kingma and Ba [2014] for both networks and applied early stopping only for $\varphi_{t,\eta}(x_0, x_1)$ based on training loss. During inference, we solved for $p_t$ at $t = 1$ using the adaptive step-size solver Tsit5 with 100 steps. We trained the RBF metric with $\kappa = 0.5$ and $\epsilon = 0.0001$ with k-means clustering, as described in Appendix C. For this experiment, we enforce stronger bending by using $(\tilde{h}_\alpha(x))^8$, in the loss function $\mathcal{L}_{g_{\text{RBF}}}(\eta)$ in (11). Note that higher powers of $\tilde{h}$ ensures that regions away from the support of $\mathcal{D}$, where $\tilde{h} < 1$, are penalized even more in the geodesic objective, which highlights the role played by the biases introduced via the metric.

Our method operates in the latent space of the Stable Diffusion v1 VAE Kingma and Welling [2014], Rombach et al. [2022], except for the k-means clustering step in RBF metric pretraining, which was performed in the ambient space.

Table 5: U-Net architecture hyperparameters for unpaired image translation on AFHQ.

| | $\varphi_{t,\eta}(x_0, x_1)$ | $v_{t,\theta}(x_t)$ |
|---|---|---|
| **Channels** | 128 | 128 |
| **ResNet blocks** | 2 | 4 |
| **Channels multiple** | 1, 1 | 2, 2, 2 |
| **Heads** | 1 | 1 |
| **Heads channels** | 64 | 64 |
| **Attention resolution** | 16 | 16 |
| **Dropout** | 0 | 0.1 |
| **Batch size** | 256 | 256 |
| **Epochs** | 100 | 8k |
| **Learning rate** | 1e-4 | 1e-4 |
| **EMA-decay** | 0.9999 | 0.9999 |

**Dataset** We used the Animal Face dataset from Choi et al. [2020], adhering to the splitting predefined by dataset authors for train and validation sets, with validation treated as the test set. Standard preprocessing was applied: upsizing to 313x256, center cropping to 256x256, resizing to 128x128, and using VAE encoders for preprocessing. Finally, we computed all embeddings using pretrained Stable Diffusion v1 VAE Rombach et al. [2022]. FID Heusel et al. [2017] and LPIPS Zhang et al. [2018] were computed using the validation sets. FID was measured with respect to the cat validation set, while LPIPS Zhang et al. [2018] between pairs of source dogs and generated cats.

### D.3 Trajectory inference for single-cell data

We performed both low-dimensional and high-dimensional single-cell experiments following the setups in Tong et al. [2023a,b]. For each experiment, the single-cell datasets were partitioned by excluding an intermediary timepoint, resulting in multiple subsets. Independent models were then trained on each subset. Test metrics were calculated on the left-out marginals and averaged across all model predictions.

We employed the Adam optimizer Kingma and Ba [2014] with a learning rate of $10^{-4}$ for $\varphi_{t,\eta}(x_0, x_1)$ and the AdamW optimizer Loshchilov and Hutter [2017] with a learning rate of $10^{-3}$ and a weight decay of $10^{-5}$ for $v_{t,\theta}(x_t)$. During inference, we solved for $p_t$ at $t$ being left-out marginal using the Euler integrator for 100 steps.

We used a 90%/10% train/validation split, excluding left-out marginals from both sets. Training samples served as source and target distributions and for calculating the metrics, while validation samples were used for early stopping. We note that these settings are slightly more restrictive (and realistic) than those reported for SF$^2$M-Geo Tong et al. [2023a], where the left-out timepoint was also included in the validation set.

**Embryoid Body dataset** We used the Embryoid Body (EB) data Moon et al. [2019] preprocessed by Tong et al. [2020], focusing on the first five whitened dimensions. The dataset, consisting of five time points over 30 days, was used to train separate models across the full-time scale, each time leaving out one of the time points 1, 2, or 3.

**Cite and Multi datasets** We utilized the Cite and Multi datasets from the Multimodal Single-cell Integration Challenge at NeurIPS 2022 Lance et al. [2022], preprocessed by Tong et al. [2023a]. These datasets include single-cell measurements from CD4+ hematopoietic stem and progenitor cells for 1000 highly variable genes across four time points (days 2, 3, 4, and 7). We trained separate models each time leaving out one of the time points 3 or 4. The data was whitened only for 5-dimensional experiments.

**Multiple constraints setting** Following a similar approach to Neklyudov et al. [2023b], we modified our sampling procedure to interpolate between two intermediate dataset marginals, with neural network parameters $\eta$ shared across timesteps. The interpolation is defined as:

$$x_t = \frac{t_{i+1} - t}{t_{i+1} - t_i} x_{t_i} + \frac{t - t_i}{t_{i+1} - t_i} x_{t_{i+1}} + \left( 1 - \left( \frac{t_{i+1} - t}{t_{i+1} - t_i} \right)^2 - \left( \frac{t - t_i}{t_{i+1} - t_i} \right)^2 \right) \varphi_{t,\eta}(x_0, x_1). \quad (20)$$

**Low-Dimensional experiments** We parameterized both $\varphi_{t,\eta}(x_0, x_1)$ and $v_{t,\theta}(x_t)$ as 3-layer MLP networks with a width of 64 and SeLU activation. In the LAND metric, we used $\sigma = 0.125$ and $\epsilon = 0.001$ for all EB sets and the first leave-out time point in the Cite and Multi datasets. For the second leave-out time point in Cite and Multi, we used $\sigma = 0.25$ and $\epsilon = 0.001$.

**High-Dimensional experiments** Both $\varphi_{t,\eta}(x_0, x_1)$ and $v_{t,\theta}(x_t)$ were parameterized as 3-layer MLP networks with a width of 1024 and SeLU activation. We set $\kappa = 1.5$ and $\epsilon$ to be the complement of the final metric pretraining loss to maintain consistent regularization across datasets and leave-out timesteps.

**Baselines** We reproduced the results reported by Neklyudov et al. [2023b] to ensure consistent reporting of standard deviations and the same versions of EB dataset used across all experiments. The standard deviations were calculated across all leave-out timesteps and seeds for each dataset and dimension. We used the provided code and hyperparameters for training, averaging the results across 5 seeds.

# E    Supplementary Single-Cell Reconstruction

We report additional results for single-cell reconstruction using 50 principal components in Table 6. Again, we note that OT-MFM performs strongly, and it is marginally surpassed only by SF$^2$ on Multi(50D) (results have not been reproduced). Two important remarks are in order. First, the baseline SF$^2$ M-Geo leverages a geodesic cost in the formulation of the Optimal Transport coupling, which enforces similar biases to OT-MFM. In fact, as argued in Section 4, OT-MFM can similarly consider data-aware costs in the formulation of the optimal transport coupling. In this work though, we wanted to focus on the ability of interpolants to lead to meaningful matching in settings where the optimal transport coupling is agnostic of the data support. Additionally, we highlight that as we move to more realistic high-dimensional settings (100 principal components, shown in Table 3) the advantages of our framework become even more apparent.

Table 6: Wasserstein-1 distance averaged over left-out marginals ($\downarrow$ better) for 50-dim PCA representation of single-cell data for corresponding datasets. Results are averaged over 5 independent runs.

| Method | Cite (50D) | Multi (50D) |
|---|---|---|
| SF$^2$ M-Geo | $38.524 \pm 0.293$ | $\mathbf{44.795 \pm 1.911}$ |
| SF$^2$ M-Exact | $40.009 \pm 0.783$ | $45.337 \pm 2.833$ |
| OT-CFM | $38.756 \pm 0.398$ | $47.576 \pm 6.622$ |
| I-CFM | $41.834 \pm 3.284$ | $49.779 \pm 4.430$ |
| WLF-UOT | $37.007 \pm 1.200$ | $46.286 \pm 5.841$ |
| WLF-SB | $39.695 \pm 1.935$ | $47.828 \pm 6.382$ |
| WLF-OT | $38.352 \pm 0.203$ | $47.890 \pm 6.492$ |
| OT-MFM$_{\text{RBF}}$ | $\mathbf{36.394 \pm 1.886}$ | $45.16 \pm 4.96$ |

# F    Supplementary Unpaired Translation Results

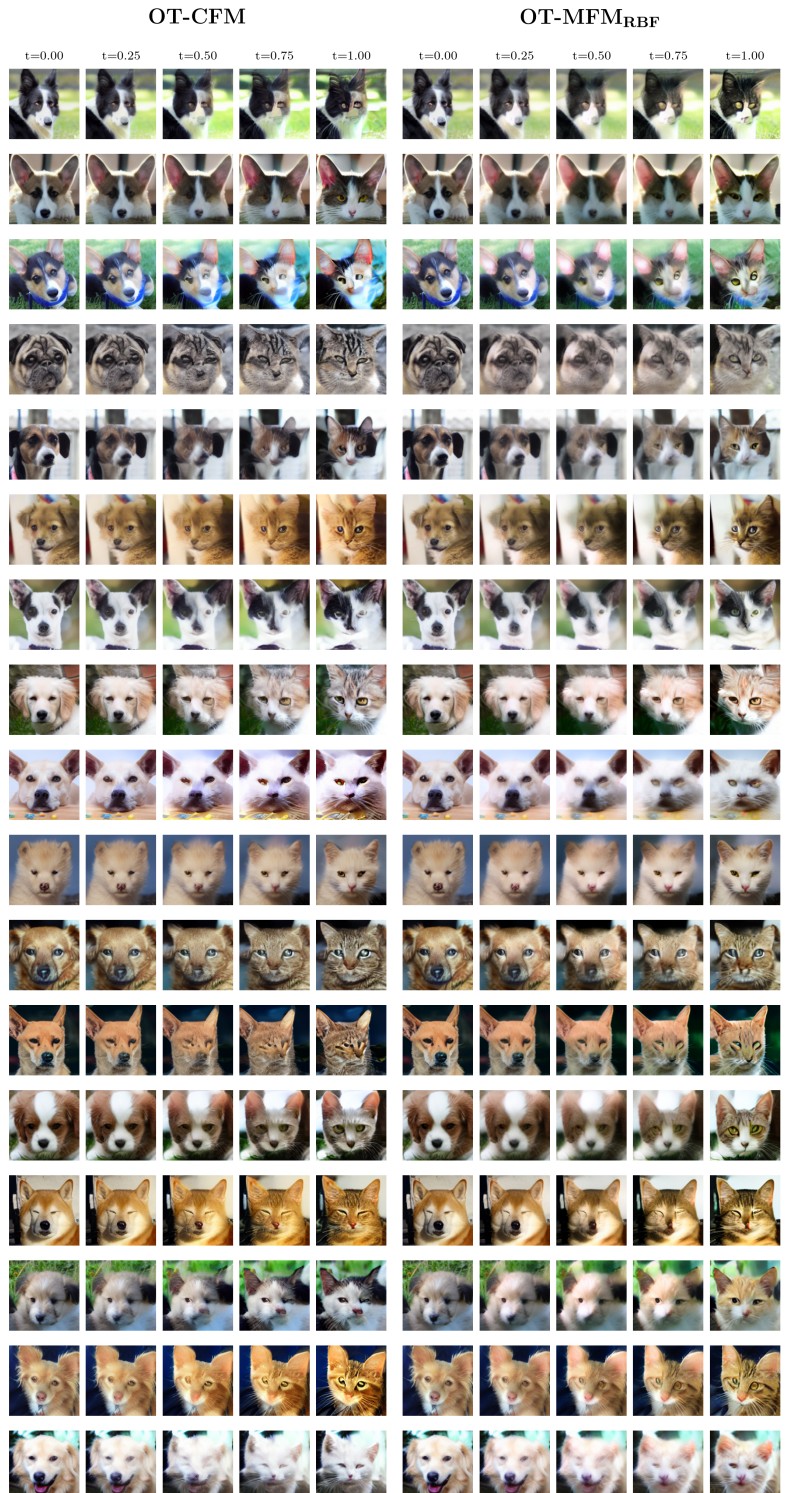

Figure 4: Additional qualitative comparison for the task of unpaired translation between OT-CFM and OT-MFM$_{\mathrm{RBF}}$.

## G    Supplementary Sphere Results

We visualize the problem setting in Figure 5.
We report the Earth Mover's Distance between
the sampled and test sets, as well as the mean
distance of the middle points of trajectories to
the sphere, to quantitatively compare CFM and
MFM on the task. MFM improves significantly
over the Euclidean baseline, CFM (Table 8). Fur-
thermore, the samples generated by MFM at
intermediate times are much closer to the un-
derlying sphere than the Euclidean counterparts
(Table 7).

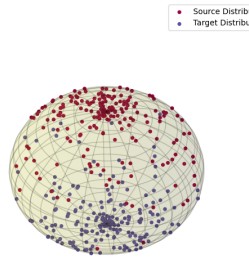

Figure 5: Problem Setup

Table 7: Mean Distance of reconstructed tra-
jectories at time $1/2$ from the sphere. Results
averaged over 5 runs.

| Method | Distance from Sphere ($\downarrow$) |
|---|---|
| OT-CFM | $0.519 \pm 0.002$ |
| OT-MFM$_{\text{LAND}}$ | $0.085 \pm 0.005$ |

Table 8: Wasserstein-1 distance between recon-
structed marginal at time $1/2$ and ground-truth.
Results averaged over 5 runs.

| Method | EMD ($\downarrow$) |
|---|---|
| OT-CFM | $0.525 \pm 0.003$ |
| OT-MFM$_{\text{LAND}}$ | $0.340 \pm 0.074$ |

## H    Supplementary LiDAR Visualizations

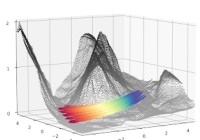 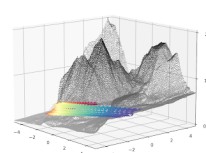 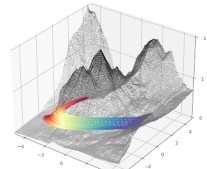 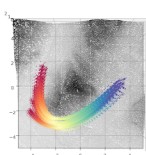

Figure 6: Supplementary Visualizations of MFM Interpolants on LiDAR

