# OpenReview forum: "Metric Flow Matching for Smooth Interpolations on the Data Manifold"
_NeurIPS.cc/2024/Conference — NeurIPS 2024 poster_

### Official Review · Reviewer_CNRE · 2024-07-04

**Soundness:** 3
**Presentation:** 3
**Contribution:** 3
**Rating:** 6
**Confidence:** 4

**Summary:**

This work proposes a metric flow matching algorithm, where interpolants are approximate geodesics learned by minimizing the kinetic energy of a data-induced Riemannian metric. This targets the trajectory inference problem, such as single-cell trajectory prediction.

**Strengths:**

* This paper clearly addresses the motivation for solving the trajectory inference problem by proposing a solution that naturally connects the recently introduced flow matching method with a data-induced Riemannian metric. The paper is well-written, with preliminaries explained compactly and effectively.
* Additionally, it thoroughly discusses the differences with Riemannian flow matching. The experiments are well compared with recent studies.

**Weaknesses:**

*Major comments*

I have two main questions and comments, which I expect to be addressed in the revised version.

* I believe the trajectory inference problem with unpaired cross-sectional data has been well addressed in "Riemannian Metric Learning via Optimal Transport (ICLR 2023)." In this work, the authors learn the Riemannian metric without introducing an auxiliary objective, but by alternatively minimizing the optimal transport objective. Your approach and this work seem to have a fundamental difference, as the objectives used for learning the Riemannian metric are different. This difference should be clearly noted, emphasizing the advantages of your approach. I hope this is included in the Introduction section with some emphasis.

* You have used L2 distance for \( c(x,y) \) in (12) due to computational issues. While this is understandable, I believe it could significantly impact the results. The two main factors affecting the resulting trajectory via flow matching are the Riemannian metric and the choice of coupling \( q \). In the original flow matching algorithm, the quality of the resulting generation was more important than the intermediate trajectory. However, in this study, the trajectory itself is the objective, making it crucial to carefully choose these two factors.
In the current version of the paper, this issue seems to be mentioned very briefly and almost as if it were insignificant. It appears necessary to include a more detailed discussion and address this point more thoroughly.

*Minor comments*

I appreciate comprehensive reviews in section 6. There are a few missing references, and I sugges to include them.
* Geometry-aware generative models:
   - Learning geometry-preserving latent spaces: Regularized Autoencoders for Isometric Representation Learning (ICLR 2022) and Geometric Autoencoders--What You See is What You Decode (ICML 2023),
   - Learning data manifolds and latent spaces by exploiting ambient space metrics: A statistical manifold framework for point cloud data (ICML 2022)'' and Geometrically regularized autoencoders for non-Euclidean data (ICLR 2023)
   - geometry-based regularization for learning accurate manifolds: Neighborhood Reconstructing Autoencoders (NeurIPS 2021) and On explicit curvature regularization in deep generative models (TAG-ML 2023).

**Questions:**

* Both $g_{RBF}$ and $g_{LAND}$ capture data density and ensure that the interpolants lie within the data support. In principle, this algorithm appears to be designed to ensure that inferred trajectories lie within the data support. Are there other trajectory inference approaches where algorithms essentially do the same thing (i.e., identify trajectories that mostly lie within the data support and connect two data distributions)? If so, is the advantage of your method that it can provide directional information when using a general Riemannian metric? Can a 'good' Riemannian metric be defined for trajectory inference that includes direction? How does this compare to ``Riemannian Metric Learning via Optimal Transport (ICLR 2023)''?

**Limitations:**

Yes

---

> ### Author Rebuttal · Authors · 2024-08-06
>
> We would like to thank the reviewer for their time and thoughtful review of our work, which gave us an opportunity to improve our work significantly. We are glad to hear that the reviewer found our work “well-written” and that it “proposes a solution that naturally connects the recently introduced flow matching method with a data-induced Riemannian metric” with “experiments that are well compared with recent studies”. We first refer to the **general rebuttal**, where we have provided additional experiments and visualizations (see the 1 page pdf). Below, we address the key clarification points and questions raised in the review one by one.
>
>
> ### Weaknesses
>
> > _I believe the trajectory inference problem with unpaired cross-sectional data has been well addressed in "Riemannian Metric Learning via Optimal Transport (ICLR 2023)." (..) Your approach and this work seem to have a fundamental difference, as the objectives used for learning the Riemannian metric are different. This difference should be clearly noted, emphasizing the advantages of your approach._
>
> We appreciate the reviewer for pointing out this highly relevant work. We have already added a citation to Scarvelis & Solomon 2023 work in a revised version of the paper. We will further comment extensively on key differences: (i) the objectives being learned and (ii) the fact that we first optimize the paths and then match the vector field rather than regularizing directly the vector field (as in their Eq. (7)). Crucially, since MFM relies on flow matching, our framework does not require simulations during training when tackling the trajectory inference problem, which is an advantage compared to the work of Scarvelis and Solomon.
> > _You have used L2 distance for ( c(x,y) ) in (12) due to computational issues. While this is understandable, I believe it could significantly impact the results. The two main factors affecting the resulting trajectory via flow matching are the Riemannian metric and the choice of coupling ( q ). (...)  It appears necessary to include a more detailed discussion and address this point more thoroughly._
>
> We appreciate the reviewer's concern. It is correct that the quality of the matching is given by both the choice of a coupling $q$—that is used to sample the boundary points $x_0$ and $x_1$—and the choice of the interpolants connecting such boundary points. First, we note that choosing a non-Euclidean cost for the OT objective is an interesting and non-trivial problem on its own. For arbitrary Riemannian metrics it is often unlikely to hope for a closed form expression to the metric-induced distance and thus OT cost. As a result, this requires us to simulate all pairs of matching to compute $c(x,y)$ which is computationally prohibitive from a practical standpoint. Furthermore, we argue an advantage of our framework is showing that even when we fix the cost to be Euclidean for the OT coupling, by learning data-dependent paths we can improve upon the Euclidean baseline.
> To further decouple the impact of the interpolants from the choice of the coupling, we conducted further experiments—found in our 1 pg global response PDF—comparing both CFM and MFM on the arch dataset and the single-cell ones, where we chose the coupling q to simply be the independent one. We agree with the reviewer that aspects of this discussion need further clarifications and we will include a larger section to explicitly outline our motivation and claims, along with the addition of the new experiments using independent coupling.
>
>
> ### Minor comments
> Thanks for sharing these works, we will add relevant citations in the revised version of our paper.
>
> ### Questions
> > _Both $g\_{\rm RBF}$ and $g\_{\rm LAND}$ capture data density and ensure that the interpolants lie within the data support. In principle, this algorithm appears to be designed to ensure that inferred trajectories lie within the data support. Are there other trajectory inference approaches where algorithms essentially do the same thing (i.e., identify trajectories that mostly lie within the data support and connect two data distributions)?_
>
>
> Regarding the choice of the metrics, we adopted those that seem to be the easiest/most efficient ones to show versatility of our framework and importantly reduce any computational overhead compared to the Euclidean FM baseline. Crucially though, we also wanted to highlight how our framework can, in principle, be applied beyond trajectory inference (as for the image interpolation problem), which is why we did not go overly-specific on the choice of the metric based on trajectories. However we believe this to be an interesting direction, i.e. how to choose task-dependent metrics to further improve the MFM framework. We hope this can be addressed in future work by us and the community more generally.
>
> ### Final comment
>
> We hope that our responses here in conjunction with the general rebuttal and the additional experiments help answer the great questions raised by the reviewer. We politely encourage the reviewer to continue asking more questions or if possible consider a fresher evaluation of our paper with a potential score upgrade.

---

> > ### Comment · Reviewer_CNRE · 2024-08-09
> > **Responses**
> >
> > Thank you for addressing all the points that I raised. I believe the clarity of the paper will increase, and I am therefore inclined to raise the score to accept.

---

> > > ### Author Response · Authors · 2024-08-09
> > > **Re: Response**
> > >
> > > We thank the reviewer for their time and engaging with us in the rebuttal. We are glad that the reviewer has found our rebuttal to increase the clarity of the paper and we hope the reviewer can also upgrade their score as they mention in their rebuttal response. We are also more than happy to answer any lingering questions the reviewer has, please let us know!

---

### Official Review · Reviewer_q2sv · 2024-07-04

**Soundness:** 4
**Presentation:** 4
**Contribution:** 3
**Rating:** 7
**Confidence:** 4

**Summary:**

This paper proposes metric flow matching, a variant of conditional flow matching where the interpolated distributions lie on the data manifold. They first learn trajectories between $p_0$ and $p_1$ which minimize a data-dependent kinetic energy, then use these trajectories instead of straight lines for conditional flow matching. The model is applied to single-cell trajectory prediction, where performance is better than the non-metric flow matching baseline.

**Strengths:**

The paper is very well written, with great attention to detail. Everything is easy to follow, including the more technical details. The authors address an interesting problem (how to get more interpretable and useful flow matching trajectories) which has received surprisingly little attention given the current popularity of flow matching. They find a suitable application in single-cell trajectory prediction and it may improve the quality of image translation models (it's hard to be confident since the image experiments are limited). To my knowledge the work is an original contribution and a useful addition to the generative modelling community.

**Weaknesses:**

I have some minor complaints:
1. It seems that the dataset $\mathcal{D}$ is the concatenation of samples from both $p_0$ and $p_1$ but this is not clearly stated
2. There is no justification of the use of diagonal metrics except that it reduces computation cost. Surely there must be some downside? Is it possible to compare the performance to non-diagonal metrics in the 2d example?
3. It's not clear how the OT is implemented. Is it a batchwise scheme? Does this introduce a bias? If not batchwise, is it scalable?
4. In tables 3 and 4, the MFM results are bolded. While the mean value is indeed the lowest, we cannot be confident that the true value is lower than some of the other values (e.g., WLF-UOT) with the stated error bounds. While I understand that this is common practice, I personally find it unscientific to report MFM values in bold and not others which could plausibly be lower than the MFM values
5. Line 289: I don't agree that the interpolations are better semantically. If there is a difference, it is too small to be able to make such a subjective claim

**Questions:**

1. Line 125: "regular" means smooth in some way? Can you be more specific?
2. Why is it preferable to use LAND for lower dimensions?
3. Am I correct that the LIDAR data is just an example of a 2d manifold embedded in 3d? I found it confusing at first since from fig 2 I thought it was encouraged to follow low-altitude trajectories and that maybe the height was being used to inform the metric. Perhaps you can chose a different visualization/update the description to avoid this confusion

**Limitations:**

The only limitation given is that the data must be embedded in Euclidean space. I am sure the authors can think of others, such as using a diagonal metric, and whatever trade-offs are involved in the OT scheme they use (see Weaknesses above)

---

> ### Author Rebuttal · Authors · 2024-08-06
>
> We thank the reviewer for their time and positive appraisal of our work. We are thrilled that the reviewer viewed our work to be “well-written, with great attention to detail” and that we “address an interesting problem” and is an “original contribution” that is a “useful addition to the generative modelling community”. We now provide responses to the main questions raised by the reviewer.
>
>  ### Weaknesses
>
> > _1. It seems that the dataset $\mathcal{D}$ is the concatenation of samples from both $𝑝_0$ and $p_1$  but this is not clearly stated_
>
> Yes that is correct—we will make this more explicit, since we work in a “generalized” trajectory inference setup, we think of samples from $p\_0$ and $p\_1$ as samples from the same dynamical system evaluated at different times so that the manifold we refer to is indeed the one spanned by the trajectories.
> > _2. There is no justification of the use of diagonal metrics except that it reduces computation cost. Surely there must be some downside? Is it possible to compare the performance to non-diagonal metrics in the 2d example?_
>
> That's a great question! Working with diagonal matrices, may potentially have some expressive power limitations in high dimensions, specifically for settings where capturing a precise underlying metric is crucial—note that in the applications considered in our submission it is hard to identify a ground-truth metric. In general though, there are theoretical reasons as to why our choice is not particularly limiting; for example, in the 2D case any metric is actually locally conformally flat (follows from the existence of isothermal coordinates). Accordingly, for 2D manifolds, one would not lose expressive power by actually taking the matrix $\mathbf{G}$ to be a multiple of the identity pointwise. Nonetheless, our framework is not constrained to diagonal metrics and more general cases are indeed possible. To further support this point, we refer to the additional experiments shared in the 1 page pdf and **general rebuttal**, where we showed how intermediate samples generated using MFM are much closer to the underlying lower dimensional manifold (a sphere) than the ones generated using CFM (see Table 4 and Figures 1,2,3). Crucially, MFM only leverages the diagonal LAND metric defined in the ambient space based on samples and is never given information about the sphere.
> > _3. It's not clear how the OT is implemented. Is it a batchwise scheme? Does this introduce a bias? If not batchwise, is it scalable?_
>
> Our method relies on using mini-batch OT which has been a standard tool in the generative modeling / Machine Learning literature (see for example Tong et al., 2023 b,c). In this case we can easily trade off the cost of OT with the batch size and in practice this does not create a severe overhead in training our models. We find that OT helps improve training stability and leads to better empirical performance. We note however that MFM with independent coupling still surpasses CFM with independent coupling, as reported in our global 1 pg response where we compare CFM and MFM without OT (Table 1,2,3).
>
>
> > _4. In tables 3 and 4, the MFM results are bolded. (..) I personally find it unscientific to report MFM values in bold and not others which could plausibly be lower than the MFM values_
>
> Thank you for your feedback. We agree with your observation and will remove bold numbers from Tables 3 and 4.
>
> > _5. Line 289: I don't agree that the interpolations are better semantically. If there is a difference, it is too small to be able to make such a subjective claim_
>
> We again agree that this may be entering subjective territory a little, so we refine our statement and claims.
>
>
> ### Questions
>
> > _1. Line 125: "regular" means smooth in some way? Can you be more specific?_
>
>
> Yes, it does mean smooth in some way. To improve clarity, when citing the paper of Ambrosio et al we refer to their explicit statement, where assumptions are stated clearly.
>
> > _2. Why is it preferable to use LAND for lower dimensions?_
>
> That's an excellent question. We empirically found LAND to be marginally better in lower dimensions, which we believe to be reasonable given that in this setting we are building the metric using all samples directly without any clustering being involved beforehand.
> > _3. Am I correct that the LIDAR data is just an example of a 2d manifold embedded in 3d? (...)_
>
> Yes that is correct, as in we do not provide height information for the metric since we wanted to test MFM in a setting where the metric was built agnostic of the downstream application. To address your concerns, in the additional 1 page pdf we have provided more views which we will add in the appendix of the revised version to improve clarity.
>
> > _The only limitation given is that the data must be embedded in Euclidean space. I am sure the authors can think of others, such as using a diagonal metric, and whatever trade-offs are involved in the OT scheme they use (see Weaknesses above)_
>
>
>
> Thank you for the feedback. We have now also added a line mentioning the trade-offs associated with the OT scheme. In light of our previous comment, we do not see any serious weakness with using diagonal metrics and in general the framework does not require one.
>
> ### Final comment
>
> We hope we have addressed the main concerns of the reviewer, and that the addition of experiments of MFM without OT and MFM on a sphere have further improved the strength of our submission. We hope that our answers enable the reviewer to continue endorsing our paper and potentially even upgrading the score if the reviewer deems it. We are also more than happy to engage in any further discussion.

---

> > ### Comment · Reviewer_q2sv · 2024-08-12
> >
> > Thank you for your detailed rebuttal. I believe you've answered all my questions. Since most of my comments were minor details, my overall judgement of the paper has not changed, so I will leave my score as is.

---

> > > ### Author Response · Authors · 2024-08-13
> > > **Thank you**
> > >
> > > We thank the reviewer for engaging with us, for the detailed and valuable feedback, and finally for endorsing our submission.

---

### Official Review · Reviewer_mgNS · 2024-07-12

**Soundness:** 4
**Presentation:** 4
**Contribution:** 3
**Rating:** 6
**Confidence:** 5

**Summary:**

The authors introduce a method for trajectory inference based on conditional flow matching (CFM) that takes a formulation of the trajectories using Riemannian geometry. The Riemannian metric is built following the manifold hypothesis and prior work, resulting in a flow matching with a data-dependent metric encouraging trajectories to stay close to the so-called data manifold. Furthermore, the sampling distribution used in the optimisation of the objective function is improved with optimal transport concepts, properly matching the joint distribution of interpolant endpoints to the main task of interest: single-cell dynamics.

**Strengths:**

The paper is very well written and easy to follow. Its main originality lies in the use of a data-dependent Riemannian metric with conditional flow matching to learn trajectories between distributions which are discouraged to stray from the data manifold. The elegant incorporation of a data-dependent metric means the method works well without excessively requiring expert knowledge about the problem domain and an appropriate metric to be chosen.

**Weaknesses:**

The technical contributions allow CFMs to be employed with a Riemannian metric and without deep understanding of an appropriate metric to be used for the task of interest. However, most of the work to get there was already proposed before. In particular, beyond Eq. (4) followed by then adapting all distances to be Riemannian (as required by the problem setting), I see little "on top" of the work by Arvanitidis et al. (2016, 2021) and Tong et al. (2023b).

Given the technical contribution builds little on top of existing work, and taking into account question 2 below regarding the image translation experiment, I believe the paper showed limited applicability beyond single-cell dynamics. I expected a stronger experimental setting showing applicability in more problems where adopting a Riemannian metric is the advantageous choice.

**Questions:**

1. Taking into account the weakness mentioned above regarding novelty: could the authors clarify their contributions beyond Eq. (4)?
2. I understand that in other domains, e.g. single-cell data, meaningfullness of trajectories might be more obvious to define, but I do not understand the claim about meaningfullness of trajectories between pictures of cats and dogs. Especially after looking at additional results in the appendix, I believe the setting is fundamentally flawed to evaluate the proposed model, with both models being similarly bad in many cases. What is the expected "good" case? Could the authors clarify their motivations for using this specific example? (By the way, the 6th row of Fig. 4 in the Appendix does not seem to be the same interpolants in both cases).
3. Can a comparison with RFM be made in a setting where the underlying "true" appropriate metric is known? The Arch data set or some variant in hyperbolic geometry seem to be interesting cases for me.

Minor comments:
- Why are there different citations for the manifold hypothesis? At first (introduction), a set of papers are cited, but then later in Section 3.1, another paper is cited.
- [l. 160] "boundary conditions": at this point I was defensively looking for some boundary conditions I might have missed, but later noticed the authors likely meant the endpoints $x_0$ and $x_1$. Is that the case? Could this be clarified at this at this point when first mentioning "boundary conditions"?

**Limitations:**

The authors have sufficiently addressed the limitations in their work.

---

> ### Author Rebuttal · Authors · 2024-08-06
>
> We thank the reviewer for their detailed feedback and constructive comments, which gave us an opportunity to improve our work significantly. We are pleased to see that the reviewer found our work “very well written and easy to follow” and that our idea is “elegant” and that “the method works well without excessively requiring expert knowledge”.  We now address key clarification questions raised by the reviewer.
>
> ### Technical novelty
>
> > _Beyond Eq. (4)(...) I see little "on top" of the work by Arvanitidis et al. (2016, 2021) and Tong et al. (2023b)._
>
> We value the reviewer’s comments regarding the technical contribution of MFM over prior works. We kindly refer to the **general rebuttal** for a detailed discussion on the main technical contributions of MFM, while here we provide a short summary.
>
> First, we hope that drawing connections from metric learning—such as the works of Arvanitidis et al—and flow matching, is in itself an important contribution that can foster new works in this space bridging different communities further. We believe that showing the role played by the data geometry (not the ambient geometry) for generative modeling is an important research direction.
>
> More crucially, in terms of technical contributions, we believe that the novelty is not just in the parameterization adopted in (4), but also, crucially, in the optimization in Eq. (6). In fact the objective in Eq. (6) is based on evaluating a *data-dependent metric Dirichlet energy* over the paths, once we sample the boundary points $x\_0,x\_1$ according to the coupling q. A key contribution in our work is that we can learn approximate geodesics using a simulation free learning objective which we find is quite useful for downstream generative modeling applications like trajectory inference. Moreover, we believe that optimizing using a velocity-induced regularization is an essential part of our framework, along with aligning the geodesic sampling with the joint distribution q. Crucially though, our framework is not bound to use a specific metric, and we simply adopted existing ones to showcase flexibility and ease of use. We hope this addresses the reviewer’s concerns, and clarifies the technical novelty of our paper.
>
>
> ### Questions
>
> > _Taking into account the weakness mentioned above regarding novelty: could the authors clarify their contributions beyond Eq. (4)?_
>
> We have addressed this point above when it comes to technical contributions.
>
> > _I understand that in other domains, e.g. single-cell data, meaningfullness of trajectories might be more obvious to define, but I do not understand the claim about meaningfullness of trajectories between pictures of cats and dogs. (...)_
>
> We agree that meaningfulness of trajectories can be better evaluated over actual dynamical systems such as for single cell data. The experiments on unpaired image translation were meant to showcase that MFM can be on par, or in fact better, than the Euclidean baseline even on settings it was not mainly designed for. In particular, we sought to attach a visual correspondence to the learned interpolation from MFM. In contrast, one can consider another exotic interpolant which may traverse other classes—or entirely off the image manifold—in trying to go from cats->dogs. Note that to further try assessing the meaningfulness of interpolations we adopted the LPIPS metric.
>
> > _Can a comparison with RFM be made in a setting where the underlying "true" appropriate metric is known? The Arch data set or some variant in hyperbolic geometry seem to be interesting cases for me._
>
> This is a great suggestion! We address this with new experiments in our 1 pg PDF. We have extended the arch task on the 2D sphere embedded in $\mathbb{R}^3$. We have found that MFM not only improves significantly over the Euclidean baseline CFM (see Table 5 in the 1 page pdf), but crucially that the samples generated by MFM at intermediate times are much closer to the underlying sphere than the Euclidean counterpart (see Table 4 and Figures 1,2,3 in the 1 page pdf and comments in our general rebuttal). We emphasize that we manage to attain this **without** explicitly parameterizing the lower-dimensional space but simply relying on the LAND metric. Finally, we note that RFM uses the ground-truth geodesics of the *standard* metric on the sphere, meaning that in general how well RFM is able to solve trajectory inference problems on the manifold depends on how well the geodesics provided by the data-agnostic metric, resemble trajectories of the underlying system. Conversely, MFM directly learns from data and can approximately recover a curved, lower-dimensional manifold from the samples.
>
>
> **Minor comments**
>
>
>
> > _Why are there different citations for the manifold hypothesis?_
>
> This is just because we wanted to cite key papers in the main body of the paper (beyond the related work section) and the distinction (i.e. not citing all of them in both cases) is due to space constraints and formatting.
>
> > _[l. 160] "boundary conditions": at this point I was defensively looking for some boundary conditions I might have missed, but later noticed the authors likely meant the endpoints  and . Is that the case? Could this be clarified at this at this point when first mentioning "boundary conditions"?_
>
> Yes, by boundary conditions we mean that the paths recover $x\_0$ and $x\_1$ at times 0 and 1, respectively. We will add an extra sentence here to improve clarity.
>
> ### Final comment
>
> We thank the reviewer again for their review and detailed comments that helped strengthen the paper—particularly through the addition of MFM experiments on the sphere. We believe we have answered to the best of our ability all the questions here and in the general rebuttal. We hope this  allows the reviewer to consider upgrading their score if they see fit. We are also more than happy to answer any further questions.

---

> > ### Comment · Reviewer_mgNS · 2024-08-12
> >
> > Dear authors,
> >
> > I am in general positive with the rebuttal. Here are a few additional comments.
> >
> > > We emphasize that we manage to attain this without explicitly parameterizing the lower-dimensional space but simply relying on the LAND metric.
> >
> > Yes, that is indeed interesting. We must however be a bit skeptical and say that in general it is hard to assume that the data is sampled so nicely along the underlying manifold. I wonder which amount of jitter or lack of proper sampling would make the LAND metric unreliable with real data. As such, one would then put the RFM requirement of having the exact metric versus the LAND approach of relying on the data as both **desirable**, but not guaranteed to be available/satisfied. I originally meant the experiment to compare that exact thing to see how much we "sacrifice" compared to ground truth by relying purely on data, but the example provided is already showing something positive in my opinion.
> >
> > After reading the other reviews and their rebuttals, I am inclined to increase my score to Accept.

---

> > > ### Author Response · Authors · 2024-08-12
> > > **Response to official comment by Reviewer mgNS**
> > >
> > > We appreciate the reviewer taking the time to engage with us further during this rebuttal.
> > > Our goal with the new ARCH on a sphere experiment was to test whether trajectories with MFM with the LAND metric lie on the sphere *without actually parametrizing the sphere*. Given this idealized setting, we see that MFM matches our intuitions and generates samples that lie on the sphere.
> > > If the spherical inductive bias ---i.e. the exact parametrization of the sphere---was known to us apriori then RFM and MFM may coincide if the sampling of data demands it. However, note that if data is irregularly sampled on the sphere then trajectories can themselves bend on the sphere, and not obey "shortest paths" on the sphere which is a setting RFM cannot model but is handled by MFM.
> > >
> > >
> > > In general, we agree with the reviewer's point that in many practical settings data may not be sampled homogeneously on an underlying manifold, and in these cases, the LAND metric may not be optimal. We would like to note however that our MFM framework can be instantiated with any choice of metric, including a parametrized/learned one such as RBF which we found works well for higher dimensions. The key point we highlight is that any metric used in MFM is informed by the sampling of data (training set) which alleviates the need for an exact parametrization of the manifold.
> > >
> > > We thank the reviewer again for allowing us to clarify these technical aspects of MFM. We would be happy to answer any further questions the reviewer has, otherwise we politely encourage the reviewer to consider increasing their score as they originally suggested they are inclined to.

---

### Official Review · Reviewer_hBAX · 2024-07-16

**Soundness:** 2
**Presentation:** 2
**Contribution:** 1
**Rating:** 5
**Confidence:** 4

**Summary:**

This paper proposes an instantiation of flow matching where the interpolants are learned by minimizing the kinetic energy defined by a nonparametric metric defined over a set of data points (a weighted L2 norm of the velocity field). This metric is defined through another weighted normal distribution, with learnable weights that ensure the metric is at the right scale.  The interpolants are defined using another neural network and optimized to reduce the kinetic energy.

**Strengths:**

Learning a metric over a low-dimensional manifold could potentially be very interesting for high-dimensional machine learning applications.

**Weaknesses:**

### Clarity

This paper lacks clarity in its exposition, and are not sufficiently careful in stating its claims.

- The paper introduces their framework by discussing the "manifold hypothesis": that high-dim data lies in a low-dim manifold. However, the choice of metric does not induce a lower dimensional manifold, as it is induced by a simple gaussian mixture model. For some of the cellular experiments, the manifold seems to actually be found by PCA and taking the first few components to define the space of the manifold.

- The paper uses very vague terminology. For instance, there is repeated mention of a "more meaningful matching" being learned. But what is a "matching" and how do you compare between two? I believe the authors are referring to the learned time-dependent probability density at intermediate times?

### Novelty

Some of the claims about novelty are overly strong and could be more clear in precisely describing this paper's contribution of using a data-dependent metric (which is an interesting direction, I just feel the paper unnecessarily sugar coats its contribution instead of stating it objectively).

- The proposed algorithm is essentially the same as Wasserstein Lagrangian Flows and GSBM, except these two existing works actually take a further step and learn the optimal transport coupling between x0 and x1 induced by the choice of metric. In contrast, I believe the main contribution of this work lies in its choice of metric defined over a finite data set. The paper repeatedly states that it is a generalization of the existing works but in fact I think of this work as an instantiation of a subset of the existing frameworks with a particular choice of metric.

- One claim is that "MFM is simulation-free and stays relevant when geodesics lack closed form" when compared to riemannian flow matching. I think this is contrasting to the case of riemannian flow matching where the interpolants are solved with an ODE solver. However, here the geodesics are also not known in closed form and instead are solved by a neural network. I think of the proposed approach just as a time-parallel method for approximating the geodesic, and feel that the above claim is too strong. There is in fact a non-trivial optimization problem happening because MFM does not have closed-form geodesics, and the reliance on a neural network further suggests it is not as simple a procedure as this claim makes it sound like.

### Empirical validation

I feel the experimental results are messy and do not provide a coherent analysis.

For instance, part of the motivation of a data-dependent metric is to impose a more useful probability path p_t. However, the probability path p_t is determined by both the coupling q(x0, x1) and the interpolant. The use of an optimal coupling should have a very strong influence on the resulting p_t, and the choice of interpolant is not independently studied with an independent coupling.

Going through each experiment section, there are concerns regarding the setup or results:

1) The LiDAR scan.
- It is unclear what the aim of this experiment is. In the paper it is mentioned that this uses a different V_t than GSBM. [Suggestion:] However, one direct comparison that should be made is to take the same V_t as GSBM but simply replace their kinetic energy with your metric-based kinetic energy. This can help answer whether the data-dependent metric is useful for this setting.
- It is hard to tell from the visualization, but do the samples from p_t stay close to the LiDAR data points or are they "floating"? [Suggestion:] i.e., is the condition || x_t - nn(x_t) || < max_i,j || x_i - x_j || satisfied? Here x_t is sampled from the learned p_t, nn returns the nearest neighbor of x_t in the LiDAR dataset, and the right-hand-side here is the maximum distance between points in the LiDAR dataset.

2) The AFHQ experiment.
- Here the FID values are extremely high, which indicates that the model is poorly fit. Looking at values reported by GSBM and the original StarGAN v2 where this dataset was introduced, it seems reasonable to expect FID values in the range of 10-20? [Suggestion:] use the same code as existing works to reproduce their results and perform a direct comparison.
- Qualitatively looking at the samples, it doesn't seem clear which method has the better samples at t=1/2. I feel this setting doesn't showcase a non-trivial p_t since it is done in the latent space of a VAE, so we know that the samples follow a normal distribution. Given this, I'm not particularly convinced that a different interpolant provides something more interpretable. [Suggestion:] Perhaps it'd be good to visualize a 2D PCA of the trajectories?
- Again, here I feel the role of the interpolant is heavily diminished when compared to the role of the optimal coupling. I believe LPIPS is computed based on pairs of (x0, x1), and that the values are extremely close seems to suggest the following: the choice of interpolant does not (or has very little) influence on the learned coupling. Since the goal of unpaired translation is to learn how to translate, it seems one should not be using the optimal coupling for training. [Suggestion:] test different interpolants while using the independent coupling.

3) The single-cell experiment.
- Here the setup is given K time points, while one time point is missing. However, we know that by construction, the probability density at this missing time point only depends on the two closest time points. [Suggestion:] Does training on all K-1 time points actually offer an improvement over training just over the nearest time points?
- Furthermore, there is the issue with the choice of coupling influencing the results. [Suggestion:] Testing different interpolants while using the independent coupling would be good to have here.
- When the optimal coupling is given, is there even a need to learn the flow? It seems this experiments can be solved by sampling pairs (x0, x1) from the coupling q and using the interpolant x_t. [Suggestion:] Does learning a flow actually provide any improvements on top of this baseline?
- I am a bit confused regarding the different dimensions (number of components used from PCA), since the initial motivation of this work is to learn the low-dimensional manifold. I feel both this and the high-dimensional image setting could be set up where the metric directly influences the dimension of the flow model, which would significantly help support the initial justification.

**Questions:**

- How effective is the interpolant when no optimal coupling is used, especially for multimodal data distributions?
- Does learning a full flow matching model outperform just using the L2 optimal coupling and using the interpolant to create x_t samples?
- How does the framework work when you have multiple time snapshots? Is the metric time-dependent (by depending on different snapshots given time) or do you fit it to a single fixed-time snapshot?
- This part confuses me: to justify the metric, it is described that p_t "stays close" to a reference dataset D, however, the evaluation settings use K separate data distributions (at K different time values). How do you determine what D is?
- Given two disjoint distributions separated apart, does the metric still learn some non-trivial p_t between them (or does the metric degenerate to the Euclidean one when far away from data)?
- Can this framework be adapted to learn a low dimensional manifold (rather than just a metric)?

**Limitations:**

There is a computational limitation when using larger training set sizes, as the majority of machine learning datasets are now in the order of millions and billions. It does not seem feasible to use a nonparametric method such as the metric proposed here.

---

> ### Author Rebuttal · Authors · 2024-08-06
>
> We would like to thank the reviewer for their detailed feedback and constructive comments, which gave us an opportunity to improve our work significantly. We are glad to hear that the main thrust of our work could be very “interesting” for higher dimensional ML applications. We kindly point the reviewer to our **general rebuttal** and 1 page pdf, which includes several new ablations and additional results. In this rebuttal, we address the main concerns of the reviewer: (i) Clarity on metric learning vs manifold learning; (ii) Novelty; (iii) Experiments using independent coupling.
>
> Below the rebuttal, i.e. in the following comments, we tackle any standing point from the review.
>
> ### Metric learning vs Manifold learning
>
> > _The paper introduces their framework by discussing the "manifold hypothesis": that high-dim data lies in a low-dim manifold. However, the choice of metric does not induce a lower dimensional manifold._
>
> The reviewer makes a pertinent observation that the choice of metric in MFM does not induce a lower dimensional manifold. To clarify why this is **intended**, given our data-points, we are interested in defining a notion of path between samples $x\_0$ and $x\_1$ that preserve the underlying geometry. There are now two equivalent ways to proceed: (i) One can find explicit coordinate representations of the underlying manifold and build the path through these coordinates (*manifold learning*); (ii) Define the path in the ambient space but change the geometry of the ambient space so that regions away from the samples (and hence the manifold) are highly penalized (*metric learning*). In practice we achieve the same result, which is having paths bending according to where the samples are. By leveraging the metric learning approach (ii), we do not need to prescribe a specific lower dimension or find a coordinate representation.
>
> To support our point further, we have followed the suggestion of reviewer mgNS and run MFM on the arch task when the samples belong to the 2D sphere in $\mathbb{R}^3$. We have found that MFM not only improves significantly over the Euclidean baseline CFM, but crucially that the samples generated by MFM at intermediate times are much closer to the underlying sphere than the Euclidean counterpart (see Table 4 in the 1 page pdf and our **general rebuttal**). We emphasize that we manage to attain this **without** explicitly parameterizing the lower-dimensional space but simply relying on the LAND metric. We hope this provides you with strong empirical evidence as to why the metric approach can help us design flows that stay close to a lower-dimensional manifold even when we do not know it.
>
> ### Novelty
>
> > _The proposed algorithm is essentially the same as Wasserstein Lagrangian Flows and GSBM (…)_
>
> We value the reviewers opinion about the similarities between our proposed MFM and WLF and GSBM.  We would like to gently push back against the reviewers assertion that these methods take a further step than MFM. In particular, we disagree with the assertion that WLF and GSBM learn an OT coupling **induced by the choice of the metric**, since no Riemannian metric (of any kind) is proposed or studied. As such, we argue WLF and GSBM cannot be considered generalizations of MFM despite their similarities.
>
> We kindly refer to the **general rebuttal** where we have provided details on technical differences between MFM and WLF and GSBM.
>
>
> > _One claim is that "MFM is simulation-free and stays relevant when geodesics lack closed form" when compared to riemannian flow matching. (...)._
>
> We appreciate the reviewers comment regarding the simulation free nature of the approximate geodesics learned in MFM. At present, there is no computational method that can find exact geodesics without simulations and we certainly do not hope to claim that MFM exactly solves these. In contrast, we believe that the strength of MFM lies in the fact that the problem of trajectory inference benefits from approximate geodesics that can easily be plugged in a conditional flow matching framework in a computationally efficient manner—i.e. they are simulation free. We further highlight that such approximate geodesics can be found thanks to our proposed objective Eq. 6, i.e.
>
> $$
> \mathcal{L}\_g(\eta) = \mathbb{E}\_{(x\_0,x\_1)\sim q,t} \left[(\dot{x}\_{t,\eta})^\top \mathbf{G}(x\_{t,\eta};\mathcal{D})\dot{x}\_{t,\eta}\right]
> $$
>
>
> where we minimize the Dirichlet energy of the path—whose optimum is in fact a geodesic. We point to the general rebuttal and to the 1 page pdf, where we showed how MFM manages to generate a flow whose intermediate samples are close to the lower-dimensional manifold by learning approximate geodesics of LAND through Eq. (6).  We hope the reviewer may now agree that our empirical evidence enables us to claim both technical and empirical novelty of using learned (simulation-free) geodesics for trajectory inference.
>
> ### Experiments using independent coupling
>
> > _(...) The choice of interpolant is not independently studied with an independent coupling._
>
> We thank the reviewer for this suggestion. In the **general rebuttal** we have provided additional experiments comparing CFM with independent coupling, i.e. I-CFM, and MFM with independent coupling, i.e. I-MFM, on the arch task and the single cell RNA datasets (in both 5D and 100D). We briefly summarize here the main takeaways:
> - I-MFM generally surpasses I-CFM.
> - We further stress that, differently from CFM, MFM also uses the coupling q in the first stage where it optimizes paths based on the metric, which justifies why using Optimal Transport for the coupling is even more beneficial for MFM than CFM.
>
> We hope that in light of the new experiments, the reviewer can see that the benefits of MFM are not just due to the choice of the coupling, since I-MFM and OT-MFM both surpass their respective Euclidean counterparts I-CFM and OT-CFM.

---

> ### Author Response · Authors · 2024-08-07
> **Rebuttal (2/3)**
>
> In these comments we now address any standing question/concern.
> > _For some of the cellular experiments, the manifold seems to actually be found by PCA_
>
> In single-cell data, current best practices suggest performing non-linear dimensionality reduction on the top 10-100 principle components. We follow this standard approach, learning a non-linear metric on the top 5/100 PCs, please see [2] and [3] below.
> > _The paper uses very vague terminology. For instance, there is repeated mention of a "more meaningful matching"_
>
> We clarify that the term matching (introduced in Score Matching and Flow Matching [Song et. al 2019, Lipman et. al 2023]) refers approximating a target distribution $p_1$ with a learned one $p_{\theta}$ via a regression objective over vector fields—i.e. $|| v\_{\theta}(x\_t, t) - u\_t(x\_t | x\_0, x\_1)||\_2$. For trajectory inference tasks “more meaningful matching” refers to learning a matching that better respects the task description, meaning that the reconstructed trajectories replicate those generated by the underlying physical system.
>
> We hope that our answer here addresses the reviewer's comment and we will update the main text to include these clarifications.
>
> **1. Lidar scan**
> > _It is unclear what the aim of this experiment is (...)_
>
> We thank the reviewer for their thoughtful suggestions. The primary purpose of our Lidar experiments is to simply contrast MFM to the Euclidean baseline (OT-CFM) and visually illustrate how the metric can affect the learned paths. While using more task-specific potentials—e.g. that account for the height information—as done in GSBM, is possible within our framework, we believe that this goes beyond the goal of this experiment which serves to highlight the benefit of incorporating data geometry **without** providing additional details from the downstream task.
>
> To address the reviewer's concern about the faithfulness of $p\_t$ to the LiDAR data, in our **general rebuttal** and additional 1 page pdf, we also provide more views of the learned paths from OT-CFM and OT-MFM . The new visualizations clearly indicate that MFM paths 1.) do not intersect the manifold and 2.) bend closely around the manifold which highlights the geometric inductive bias.
>
> **2. The AFHQ experiment**
> > _Here the FID values are extremely high (...)_
>
> We acknowledge the reviewers' healthy skepticism regarding our reported FID values. We wish to highlight that other papers e.g. [4] have empirically reported FID values for interpolation of cats→dogs to be around 70. As such we believe that our values are acceptable.
> > _Qualitatively looking at the samples, it doesn't seem clear which method has the better samples at t=1/2.(...)_
>
> We agree with the reviewer’s observation. Note however that in this application the goal is not much about having better samples at the intermediate times, but having samples at $t=1$ that are more similar to those at $t=0$ as measured by the LPIPS metric, which we have reported in Table 2. In general though, the main goal of this experiment was to showcase how MFM can be used along with or instead of CFM, even for settings outside the trajectory inference task which is what was mainly designed for.
> > _Again, here I feel the role of the interpolant is heavily diminished when compared to the role of the optimal coupling.(...)_
>
> We thank the reviewer for their suggestions. As noted above and detailed in the **general rebuttal**, we have conducted experiments comparing CFM and MFM using independent coupling for both the Arch task and single cell RNA sequencing to validate that I-MFM surpasses I-CFM.
>
> **3. The single-cell experiment**
> > _Here the setup is given K time points, while one time point is missing. However, we know that by construction, the probability density at this missing time point only depends on the two closest time points(...)_
>
> This is an interesting question. In our framework, the metric $\mathbf{G}$ is constructed using two consecutive dataset marginals—based on a dataset $D\_{i,j}$, which is a concatenation of samples from these marginals (excluding $p\_{\text{out}}$). For instance, considering the scRNA dataset with densities $p\_0$, $p\_1$, $p\_2$, and $p\_3$, and excluding $p\_1$ as the left-out density, we use separate metrics $\mathbf{G}_{0,2}$ and $\mathbf{G}\_{2,3}$ for the pairs $\{p\_0, p\_2\}$ and $\{p\_2, p\_3\}$. In this regard, the definition of the metric already follows the procedure you suggested.
>
> In contrast, a *single* time-dependent interpolant network $\varphi_{t,\eta}$ and a *single* vector field networks $v\_{t, \theta}$ are used for all time-steps ($t\_0$, $t\_2$ and $t\_3$) to ensure continuity of trajectories across different times. This also ensures that we follow standard procedures adopted by the baselines reported in Tables 3 and 4.
> > _Issue with the choice of coupling (...)_
>
> We have addressed this point above and in the **general rebuttal**. Once again, we highlight that I-MFM is better than I-CFM.

---

> ### Author Response · Authors · 2024-08-07
> **Rebuttal (3/3)**
>
> > _When the optimal coupling is given, is there even a need to learn the flow?_
>
> That's an interesting suggestion! Unfortunately, single cell RNA is known to have a destructive generative process which means that different time marginals can contain varying numbers of *unpaired* observations. Operationally, this means there is no 1-1 correspondence between populations at different times. Consequently, evolving the particles along a path $x\_t$ may be an ill-posed problem and instead, we must assess the probability path $p\_t$. We achieve that by taking the pushforward of $p_0$ using the flow generated by the vector field $v_\theta$ learned using flow matching. We understand how this technical point may not have been clear given the complex nature of single cell data and now hope the reviewer agrees that their suggestion, while interesting, cannot be employed in this particular experiment.
>
> > _I am a bit confused regarding the different dimensions (number of components used from PCA), since the initial motivation of this work is to learn the low-dimensional manifold. (...)_
>
> We appreciate the reviewer's concern. We politely point out that trajectory inference using PCA of single-cell data is not an innovation of our work but is a large subfield in its own right (see [2],[3]). Practitioners often resort to PCA as actual single cell data is both very high dimensional and noisy which provides a computationally intractable domain for learning trajectories prior to simulation free generative modeling. In this context we argue our reported benchmarks are standard in the literature. We also point again to our new experiments on the sphere (see **general rebuttal**), highlighting the ability of MFM to learn trajectories that stay closer to an unknown, underlying manifold.
>
>
> ### Questions
>
> > _How effective is the interpolant when no optimal coupling is used, especially for multimodal data distributions?_
>
> Please see our response/additional experiments on using independent coupling.
>
>
> > _Does learning a full flow matching model outperform just using the L2 optimal coupling and using the interpolant to create x_t samples?_
>
> Please see our response above on why using just the interpolants for single cell RNA can be reductive due to the lack of a 1-1 correspondence between samples and the need to evaluate the time-evolution of the **population density**.
>
> > _How does the framework work when you have multiple time snapshots? Is the metric time-dependent (by depending on different snapshots given time) or do you fit it to a single fixed-time snapshot?_
>
> Please see the reply to the first point of single cell experiment setup.
>
> > _This part confuses me: to justify the metric, it is described that $p\_t$ "stays close" to a reference dataset D, however, the evaluation settings use K separate data distributions (at K different time values). How do you determine what D is?_
>
> Please see the reply to the first point of single cell experiment setup.
>
>
> > _Given two disjoint distributions separated apart, does the metric still learn some non-trivial p_t between them (or does the metric degenerate to the Euclidean one when far away from data)?_
>
> This is an interesting question and we believe that given the setting you described, if the supports of the marginal distribution is fully separated, then there is no meaningful signal for the nonlinear correction $\varphi\_{t,\eta}$ in Eq. (4) and hence the path should remain linear, perhaps after adding a penalization over the norm of the weights of the MLP $\varphi\_{t,\eta}$.
>
> > _Can this framework be adapted to learn a low dimensional manifold (rather than just a metric)?_
>
> Please see our response above on why MFM is not designed to learn a low dimensional manifold and why this is not an issue in practice (we point again to the new experiments of MFM on the sphere).
>
> ### Final comment
>
> We thank the reviewer again for their valuable feedback and great questions that enabled us to include new results that have strengthened our paper. We hope that our rebuttal addresses their questions and concerns—particularly in regard to how MFM does not need to prescribe a lower-dimensional space, as shown by the new experiments on the sphere, and the comparison with CFM using independent couplings. We kindly ask the reviewer to consider upgrading their score if the reviewer is satisfied with our responses. We are also more than happy to answer any further questions that arise.
>
> ### References
>
> [1]:  “A geometric take on metric learning”, Hauberg et al., Advances in Neural Information Processing Systems, 2012.
>
> [2]: “Best practices for single-cell analysis across modalities.”, Heumos, L., Schaar, A.C., Lance, C. et al. Nat Rev Genet (2023). https://doi.org/10.1038/s41576-023-00586-w
>
> [3]:~ https://www.sc-best-practices.org/preprocessing_visualization/dimensionality_reduction.html
>
> [4]: “Contrastive Learning for Unpaired Image-to-Image Translation” Park et al., Computer Vision-ECCV (2020)

---

> > ### Author Response · Authors · 2024-08-13
> > **Kindly awaiting more feedback**
> >
> > We thank the reviewer again for your time and feedback that allowed us to strengthen the paper with new experiments and clarifications during this important rebuttal period. As the end of the rebuttal period is fast approaching we were wondering if our answers in the rebuttal were sufficient enough to address the important concerns raised regarding 1.) clarity of our claims, 2.) the technical novelty that distinguishes our proposed approach MFM and Wasserstein Lagrangian Flows/GSBM, and 3.) the experimental evaluation. We highlight that our global response includes new ablations on the use of couplings following the great suggestions by the reviewer.
> >
> > We would be happy to engage in any further discussion that the reviewer finds pertinent, please let us know! Finally, we are very appreciative of your time and effort in this rebuttal period and hope our answers are detailed enough for the reviewer to consider a fresher evaluation of our work with a potential score upgrade if it's merited.

---

> > > ### Comment · Reviewer_hBAX · 2024-08-13
> > >
> > > Thanks for the additional ablations. I have increased my score, and would further encourage providing additional clarity and honesty in highlighting the contributions. I agree with the points raised by the authors in the rebuttal: (1) the use of a predefined non-parameteric metric is the main concept that sets this apart from existing works which only use parametric metrics, (2) however this comes at a higher computational cost compared to existing works due to (i) numerically solving individual trajectories as well as (ii) the evaluation of a non-parametric metric also involving neural network evaluations, and (3) the proposed method is not fully justified as a solution to the typical manifold hypothesis discussed in machine learning (i.e., computational cost is not dependent on the data manifold but still on the original space) and is focused purely on learning a metric in the original data space.

---

> > > > ### Author Response · Authors · 2024-08-13
> > > >
> > > > Thank you for taking the time to engage with us further during this rebuttal. We also appreciate the increase in score. However, we would like to take this last chance to clarify some misconceptions about our work you have raised.
> > > >
> > > > -   **Metric:** We would like to stress that, to the best of our knowledge, existing works do not use parametric metrics. GSBM or WLF argue in favor of potentials (that are generally not defined), but do not adopt a Riemannian formalism to assess the velocity of the path, which we believe to be what sets our work apart, independent of the choice of the metric. We note that we consider both parametric and non-parametric metrics in our work, as outlined in section 4.1 and Appendix C of the original manuscript.
> > > > -   **Cost and Complexity:**  We would like to politely note that MFM does not require numerical solving of trajectories at any stage of the training since it is a simulation free algorithm. Further, the complexity of the training is at most twice the standard CFM, as we always use the same network architecture for both training phases. At inference, MFM does not require metric meaning it is exactly as efficient as the standard CFM. All these points, make MFM efficiency comparable with GSBM and WLF.
> > > > -   **Manifold Hypothesis:**  The results from the arch experiment lifted onto a sphere (general rebuttal), suggest that MFM successfully preserves the underlying geometry without explicitly inducing a lower-dimensional manifold --- our approach maintains proximity to the manifold without the need for explicit parameterization.
> > > >
> > > > We hope this message helps to clarify the lingering concerns. We will add key points discussed in the rebuttal, including extended comparisons with GSBM and WLF, in the revised version of the manuscript.

---

> ### Comment · Reviewer_hBAX · 2024-08-13
>
> Unfortunately, the authors did not seem to understand my concerns about clarity and the paper's claims. I want to clarify that the only concern that was addressed by the rebuttal was that the choice of metric has advantages regardless of the choice of coupling used during training.
>
> **Metric:**
>
> "to the best of our knowledge, existing works do not use parametric metrics" --> To be clear, by parametric (in contrast to non-parametric) I simply meant that the metrics do not depend on a large data set. But I believe here the authors are using parametric as a synonym for learned; however, even then, **this claim is certainly false**. There has been many works on learning parametric metrics over data sets / point clouds [1,2], and many works on using latent spaces for transport problems where the metric is induced by the autoencoder, such as the DSBM and GSBM methods [3,4]. **Relatedly, it may be worth noting that the authors did not address my concern on reproducing FID: DSBM reports 14.16 and GSBM reports 12.39 FID**. Both are significantly better than the values reported by this paper (37-41), and this is especially concerning because existing works have open source implementations available. Instead, it seems the authors have deliberately chosen not to compare against *the different choices of potentials* in these works.
>
> "in favor of potentials (that are generally not defined)" --> I am not sure what is meant by not defined here, as I understand these prior works work for any potential. **Since the potentials can be arbitrarily defined, it fully includes the Lagrangian point of view of Riemannian geodesics.** As mentioned in Appendix C.1, one can always convert from the cost defined using a Riemannian metric to the regular Euclidean metric plus a potential.
>
> "do not adopt a Riemannian formalism to assess the velocity of the path" --> I believe the authors are describing how to extract a Riemannian manifold from a data set. There is significant work in this area (see the literature on constructing Riemannian metrics from point clouds e.g. [1, 2]) and is often used for LiDAR. Actually, it is worth highlighting that **my concern regarding the LiDAR experiment has also not been addressed**: compared to the GSBM experimental result which show two modes of trajectories [4; Figure 4], the results from this paper only show one mode: why is this? For me, I am not convinced that this LAND metric is as good as the choices of potential compared to existing works.
>
> **Cost and Complexity:** Here is a simple question: Given the LAND or RBF metric definition, do you have access to all geodesics in closed form? If so, then I agree the method would be simulation-free and very computationally efficient; however, from my understanding, **the answer is no. And the proposed workaround is to actually fit a parametric approximation to the geodesics.** This, as the authors also agree, have significant computational overhead and can increase costs to double the amount for CFM due to this extra step of needing to approximate geodesics with neural networks.
>
> **Manifold hypothesis:** Taking wikipedia as a proper "definition", the manifold hypothesis, if it were to be true, would imply at least two things:
>
> 1) "Machine learning models only have to fit relatively simple, low-dimensional, highly structured subspaces within their potential input space (latent manifolds)."
>
> This, as also reflected by the rebuttal, does not hold for the proposed method. The proposed method works in the original data space.
>
> 2) "Within one of these manifolds, it’s always possible to interpolate between two inputs, that is to say, morph one into another via a continuous path along which all points fall on the manifold."
>
> As noted in the review, and also agreed upon by the rebuttal response, the intermediate samples (see Figure 3 for images, and the rebuttal pdf for sphere and LiDAR) certainly do not seem to lie on the data manifold. *This is a subtle point that I raised which I believe the authors have not understood.* **If one were to actually define a manifold (along with a tangent plane and a corresponding metric on that plane), then interpolations will never leave the manifold, no matter how poorly the model/distribution is fit.** This is the key idea behind the literature on constructing Riemannian manifolds as described above, where we can fully construct a proper manifold and not just a metric in the ambient space. However, the proposed approach does not learn a subspace and due to this, we see that the points are some distance away from the manifold (shown for both sphere and LiDAR visualizations in the rebuttal pdf).
>
> [1] "Neural FIM for learning Fisher information metrics from point cloud data" Fasina et al. 2023.
>
> [2] "Approximating the Riemannian Metric from Point Clouds via Manifold Moving Least Squares" Sober et al 2020.
>
> [3] "Diffusion Schrödinger Bridge Matching" Shi et al 2023.
>
> [4] "Generalized Schrödinger Bridge Matching" Liu et al 2023.

---

> > ### Author Response · Authors · 2024-08-13
> > **Thank you for the additional comments**
> >
> > We thank the reviewer for taking the time to continue engaging with us. We believe certain aspects of our response may have added confusion rather than clarity which we apologize for. We now attempt to answer the points raised by the reviewer’s latest response.
> >
> > > "to the best of our knowledge, existing works do not use parametric metrics" …  But I believe here the authors are using parametric as a synonym for learned;
> >
> >
> > Yes, “parametric” here refers to a metric that learned via gradient based optimization. In contrast, we use non-parametric to refer to a metric that does not require learning via gradient based optimization.
> >
> > >however, even then, this claim is certainly false. There has been many works on learning parametric metrics (...) and many works on using latent spaces for transport problems where the metric is induced by the autoencoder
> >
> > We also agree with the reviewer that our work is not the first to learn a data-dependent metric. Our principal claim, that we clarify, is that MFM is the first method that employs a data-dependent (parametric or non-parametric) Riemannian metric within the context of flow matching. While it is certainly true that DSBM and GSBM have experiments that use the latent space of an auto-encoder, they do not invoke the Riemannian formalism which would have required the use of the metric to define distances, vector fields, and interpolants. As a result, we view the methodology of GSBM in its techniques to bias paths complementary to the approach presented in this paper. Indeed, there are similarities between the goals of our work and GSBM—which will be included in a larger discussion in the main paper—but crucially the key difference lies in the objective and mechanism used to bias paths. We understand how these techniques may not have been initially clear but we will use this response as a guide to further increase clarity and transparency of our novelty claims and the difference between methods.
> >
> > > the authors did not address my concern on reproducing FID: DSBM reports 14.16 and GSBM reports 12.39 FID.
> >
> > Please note we did attempt to answer this in our original rebuttal but we now include additional detail that we believe could strengthen the original response. We first highlight the 3 main reasons why our experimental setup differs from GSBM which lead to discrepancies between FID numbers between GSBM and MFM:
> > - The chief reason is that in our experiment we use the latent space of a pre-trained VAE autoencoder in StableDiffusionV1 as opposed to a more shallow VAE used in GSBM. As a result these GSBM and MFM in our current draft do generative modeling using two very different latent spaces.
> > - The GSBM codebase resizes each input image to 64 x 64 while we resized it to 128 x 128 to use our pretrained autoencoder.
> > - GSBM and DSBM operate directly on the ambient space and do not actually generate via the decoder of the VAE. In MFM as we heavily use the Riemannian metric for latent flow matching we generate using the decoder of the VAE.
> >
> > We also took this time to investigate the reviewers’ great suggestion and attempted to compute an FID using the *saved checkpoints of the UNET and VAE from GSBM*  with their provided sampling notebook. Unfortunately, we were unable to reproduce the FID of 14.16 and achieved an FID of 29.5 which is lower but closer to the one we report.
> >
> > > Instead, it seems the authors have deliberately chosen not to compare against the different choices of potentials in these works.
> >
> > The purpose of the image translation experiments was to compare CFM with MFM. In particular, we wanted to show how the **same** metric, i.e. RBF, can be fit to different applications and does not require adjustments specifics to the task. We hope the reviewer can see the merit in that. Regarding potential comparisons with GSBM for example, the setups are actually entirely different so simply comparing reported FID would not be accurate (see our previous point for more details).
> >
> >
> > > "in favor of potentials (that are generally not defined)" --> I am not sure what is meant by not defined here
> >
> > We apologize for the vague terminology. By “not defined”, we mean as in the case of GSBM that the potential has to be defined by the modeler. In our setting instead, we wanted to propose a choice of metrics e.g. LAND and RBF that could be readily applied to different downstream tasks, without explicit encoding of the task outside of using data samples.
> >
> > > Actually, it is worth highlighting that my concern regarding the LiDAR experiment has also not been addressed (...)
> >
> > We appreciate the reviewers' comments. We note that LAND in itself is not a contribution of this work but rather its use in flow matching. The main purpose of LiDAR is to visually demonstrate the impact of using a data dependent metric (in contrast to CFM) on the trajectory in flow matching. We argue that the current experiment adequately serves this goal and does not gain additional benefit from comparing to GSBM.

---

> > > ### Author Response · Authors · 2024-08-13
> > > **Thank you for additional comments (part 2)**
> > >
> > > > Cost and Complexity: Here is a simple question: Given the LAND or RBF metric definition, do you have access to all geodesics in closed form? If so, then I agree the method would be simulation-free and very computationally efficient;
> > >
> > >
> > > We politely disagree with the reviewer. Simulation-free training means that we are able to jump to any time $t$ of the interpolants vector field $u\_t (x|z)$ without simulating the path up to $<t$. **An interpolant does not need to be a geodesic for simulation free training to still hold**. Our work never sought to claim that MFM is able to extract exact geodesics in a simulation free manner but rather the learned interpolants for estimating the metric-induced velocity does not require solving a differential equation (and/or backpropagating through it) and learning requires only pointwise evaluations.
> > >
> > > > The proposed workaround is to actually fit a parametric approximation to the geodesics …
> > >
> > > Certainly, if our network successfully finds the global optimum by minimising the Dirichlet energy then we do recover a geodesic of the data manifold. The step of approximating geodesics instead of simulating the Euler Lagrange equations is a key contribution of our work that shows one can leverage benefits from metrics via simulation-free training to lead to measurable benefits for trajectory inference tasks.
> > > We disagree with the reviewer’s assessment that this cost is expensive as it can be thought of as a 1 time pre-processing step which notably is not more expensive than the original trajectory inference problem itself. We argue that the performance gains of MFM in domains such as single-cell justify this pre-processing overhead.
> > >
> > > ### Regarding the manifold hypothesis
> > >
> > >
> > > > The proposed method works in the original data space.
> > >
> > > In general, the reviewer is correct in identifying that if we care about *explicitly* parameterizing a lower-dimensional space, then one needs coordinate maps.For the sphere, the fact that the points do not lie *exactly* on the surface  is expected, considering that we are assuming we do not know the underlying manifold. In practice, for the application of trajectory inference one does not need the explicit manifold, but simply cares about reconstructing trajectories whose intermediate points are closer to the underlying manifold. Empirically, we have shown the impact of our metric with the single-cell experiments, where it is even hard to define what the ground-truth manifold is and  adopting the metric approach is favorable to finding an explicit parameterization of the lower-dimensional manifold since it requires the notion of a norm in the ambient space.
> > >
> > > We emphasize that “just a metric in the ambient space” can actually induce a non-trivial Riemann tensor and hence a “proper” Riemannian manifold. More generally, the idea of learning a distance in the ambient space rather than parameterising a lower dimensional space **is not a new approach and we are certainly not the first ones arguing for it**, but plenty of previous works (referenced in our manuscript) have adopted this angle (referred to as **metric learning**). As such, we believe our work is using this perspective in the context of generative modeling and has been substantially motivated by previous literature.

---

### Author Rebuttal · Authors · 2024-08-06

We thank the reviewers for their thoughtful feedback and constructive questions that have helped us improve the submission significantly. We are glad to see that reviewers found our work “well-written, with great attention to detail” (R q2sv), that our proposed framework is “elegant” (R mgNS), “naturally connects the recently introduced flow matching method with a data-induced Riemannian metric” (R CNRE), and “could be very interesting for higher dimensional ML applications” (R hBAX). In this general response we address two points: (1) an overview of the new experiments we ran to address shared questions raised in the reviews; (2) the technical novelty and contributions of this work.

We discuss the additional experiments we conducted during the rebuttal period. *We will refer to the attached 1 page document*.

### Additional experiments & Ablations

**Independent coupling on arch and single cell (R hBAX, R CNRE)**

To assess the impact of metric learning in MFM even without using Optimal Transport for the coupling $q$, we tested MFM with independent coupling on the Arch task and the single cell datasets (on both 5D and 100D). The results are reported in **Tables 1,2,3**  in the attached pdf, and highlight two key takeaway messages:
- I-MFM generally surpasses I-CFM

- Differently from CFM, MFM also uses the coupling q in the first stage where it optimizes paths based on the metric, which justifies why using Optimal Transport for the coupling is even more beneficial for MFM than CFM.

We hope that in light of these experiments, reviewers can see that the benefits of MFM are not just due to the choice of the coupling, since I-MFM and OT-MFM both surpass their respective Euclidean counterparts I-CFM and OT-CFM.

**Arch experiment on a explicit manifold (R hBAX, R mgNS, R q2sv)**

To assess the ability of MFM to learn trajectories that stay close to an unknown, underlying manifold, we followed the great suggestion of R mgNS and ran MFM on the arch task defined on a specific lower dimensional space, i.e. a 2D sphere embedded in $\mathbb{R}^3$. We see that MFM not only improves significantly over the Euclidean baseline CFM (see Table 5 in the attached pdf), but crucially that the samples generated by MFM at intermediate times are much closer to the underlying sphere than the Euclidean counterpart (see Table 4 and Figures 1,2,3 in the 1 page pdf). We emphasize that we managed to attain this **without** explicitly parameterizing the lower-dimensional space but simply relying on the LAND metric. We hope this provides the reviewers stronger empirical evidence as to why the metric approach can help us design flows that stay close to a lower-dimensional manifold even when we do not know it.


**LIDAR visualizations (R hBAX, R q2sv)**

To address the reviewers' concerns about the faithfulness of $p_t$ to the LiDAR data, in our additional 1 page pdf, we have provided more views of the learned paths from OT-CFM and OT-MFM (see Figure 4). The new visualizations clearly indicate that MFM paths 1.) do not intersect the manifold and 2.) bend closely around the manifold which highlights the geometric inductive bias.

### Technical novelty and contributions (R hBAX, R mgNS)

In light of some concerns about technical contributions of our submissions, we reiterate and clarify what we believe to be the key novelty of our work.

**Connecting metric learning to flow matching**

To the best of our knowledge, this is the first work that links metric learning to recent state-of-the-art generative frameworks such as flow matching and emphasizes the role played by the data geometry (*not* the ambient geometry) for generative modeling.

**Learning interpolants that approximate geodesics**

The key technical contribution of our work consists in learning stochastic interpolants that approximate geodesics of a data-dependent metric in a simulation-free manner. At present, there is no computational method that can find exact geodesics without simulations. Instead, MFM proposes to find approximate geodesics by minimizing the **metric Dirichlet energy of the path** in Eq. 6, whose minimizer is the geodesic, i.e.

$$
\mathcal{L}\_g(\eta) = \mathbb{E}\_{(x\_0,x\_1)\sim q,t} \left[(\dot{x}\_{t,\eta})^\top \mathbf{G}(x\_{t,\eta};\mathcal{D})\dot{x}\_{t,\eta}\right].
$$

To our knowledge, this is a novel objective and, importantly, can be fully decoupled from the matching objective used to learn the vector field generating the flow. Finally, our framework is general and is not bound to use a specific metric.

**Comparisons with GSBM/WLF**

In light of reviewer hBAX’s comments about similarities between MFM and WLF/GSBM—which we discuss in Section 4.2 of our paper—we comment further on **technical differences** between these frameworks. Since GSBM is an explicit matching counterpart of WLF, we focus on comparing MFM to WLF.
- (i) In contrast to WLF or GSBM, our method, MFM, separates the optimization of the paths from the matching objective. We argue this is beneficial since it avoids introducing additional challenges when learning the vector field $v\_\theta$.
- (ii) None of the Lagrangians considered in WLF account for the data distribution and/or the geometry induced by it. As such, the choice of Lagrangian always needs to be given by the _user based on overall considerations, e.g. unbalanced OT, or one needs to define a potential V_—that only depends on positions and not velocities, as discussed in 4.2. In MFM instead, we explicitly account for the empirical samples to learn the paths. If we adopt the same procedure for optimization (i.e. 2-stage learning, as clarified in (i)), then our approach can be interpreted as learning paths by minimizing a Lagrangian whose potential depends on both positions and velocities and is further dependent on the whole data distribution through the metric.

---

### Decision · Program_Chairs · 2024-09-25

**Decision:**

Accept (poster)

**Comment:**

The paper proposes to endow flow matching methods with the non-parametric LAND metric to thereby better adapt to the data distribution. Several significant critiques were raised, especially regarding the validity of the conducted experiments. The reviewers were split and did not reach a consensus. The authors are expected to take the reviewer's feedback into account when finalizing the manuscript.